## Replication

psychology/behaviour/cognition

social affect, empathy, compassion, social cognition, theory of mind, adult lifespan

**Author for correspondence:**
Julia Stietz
e-mail: julia.stietz@tu-dresden.de

# The ageing of the social mind: replicating the preservation of socio-affective and the decline of socio-cognitive processes in old age

Julia Stietz[1], Lena Pollerhoff[2], Marcel Kurtz[1],
Shu-Chen Li[2,3], Andrea M. F. Reiter[2,4] and
Philipp Kanske[1,5]

[1]Clinical Psychology and Behavioral Neuroscience, Faculty of Psychology, [2]Lifespan Developmental Neuroscience, Faculty of Psychology, and [3]Centre for Tactile Internet with Human-in-the-Loop, Faculty of Psychology, Technische Universität Dresden, Dresden, Germany
[4]Department of Child and Adolescent Psychiatry, Psychosomatics and Psychotherapy, University of Würzburg, Würzburg, Germany
[5]Max Planck Institute for Human Cognitive and Brain Sciences, Leipzig, Germany

JS, 0000-0003-4904-1850; LP, 0000-0003-4520-2266;
MK, 0000-0002-2244-6025; S-CL, 0000-0001-8409-5390;
AMFR, 0000-0002-5209-3996; PK, 0000-0003-2027-8782

Anticipating population ageing to reach a historically unprecedented level in this century and considering the public goal of promoting well-being until old age, research in many fields has started to focus on processes and factors that contribute to healthy ageing. Since human interactions have a tremendous impact on our mental and physical well-being, scientists are increasingly investigating the basic processes that enable successful social interactions such as social affect (empathy, compassion) and social cognition (Theory of Mind). However, regarding the replication crisis in psychological science it is crucial to probe the reproducibility of findings revealed by each specific method. To this end, we aimed to replicate the effect of age on empathy, compassion and Theory of Mind observed in Reiter and colleagues' study (Reiter *et al*. 2017 *Sci. Rep.* **7**, 11046 (doi:10.1038/s41598-017-10669-4)) by using the same ecologically valid paradigm in an independent sample with similar age ranges. We were able to replicate the previously observed results of a preservation or even enhancement in

socio-affective processes, but a decline in socio-cognitive processes for older adults. Our findings add to the understanding of how social affect and cognition change across the adult lifespan and may suggest targets for intervention studies aiming to foster successful social interactions and well-being until advanced old age.

# 1. Introduction

As population ageing is at its peak worldwide with changing social demographics and increasing life expectancies [1–3], it becomes more and more important to understand how we can age successfully with a healthy body and mind. While a lot of research has investigated the effects of various factors such as physical activities, diet, drug usage, non-invasive brain stimulation as well as socio-economic status to answer this question [4–9], a seminal study by Holt-Lunstad et al. [10] showed that social relationships play the most important role for healthy ageing and longevity, even more so than smoking, alcohol consumption, physical activities, as well as the body mass index and exposure to air pollution. An earlier longitudinal study also showed that social participation attenuated cognitive decline [11]. Thus, mechanisms that allow older adults (OA) to sustain an active social life could be a promising avenue to promote health and well-being until advanced old age. Thus far, most research on this topic has focused on measures such as social network size and relationship quality derived from questionnaire-based assessments. However, to engage in and successfully manage social relationships, it is important to understand the driving forces of our interaction partners' behaviour, that is, their mental states including emotions, intentions, and beliefs [12–14]. Research over the last decades has identified socio-affective and socio-cognitive routes to understanding others, that need to be differentiated in their contribution to social behaviour [15,16].

Socio-affective processes include empathy and compassion. The definition of empathy ranges from a multifaceted construct—comprising affect sharing and empathic concern as an emotional and mentalizing or perspective taking as a cognitive component of empathy [17–20]—to a very narrowly circumscribed construct—confining it to pure affect sharing [15,21]. Here we apply the later definition that specifies empathy as feeling *with* somebody [21]. More precisely empathy describes the sharing of another person's feeling while knowing that the other person's emotion is the cause of one's own feeling [21,22]. Compassion, on the other hand, is defined as a positive, caring feeling *for* somebody that is strongly linked to the motivation to help the other person [22]. Socio-cognitive processes include perspective taking also known as mentalizing or Theory of Mind (ToM), which is defined as reasoning about the mental states (e.g. thoughts, desires, feelings) of another person [23,24]. Some researchers also distinguish between an affective and cognitive component of ToM, with the former being the reasoning about emotions and the latter the reasoning about thoughts and beliefs [25]. In the current study, we mainly focus on cognitive ToM (in the following only referred to as ToM).

Socio-affective and socio-cognitive processes are not just conceptualized distinct, there is also evidence from behavioural, neuroimaging and (sub-)clinical studies that supports their independence [15,26]. Investigating empathy and ToM within the same individuals Kanske et al. [27] found no correlation between those two capacities on the behavioural and neural level. Moreover, brain areas associated with empathy encompassing the anterior insula, inferior frontal gyrus, anterior middle cingulate cortex, ventral striatum, and medial orbitofrontal cortex [28–30] mostly overlap with the salience network [31], while areas associated with ToM encompassing the temporoparietal junction, precuneus/posterior cingulate cortex, medial prefrontal cortex, superior temporal sulcus and temporal poles [28,32,33] mostly overlap with the default mode network [31]. Furthermore, research on autism, for example, points to an intact empathy, but an impaired ability to take the perspective of others in individuals with autism spectrum disorder [34–39]. Whereas research on aggressive offenders suggest an intact ToM, but a reduced capacity to emphasize with others in these individuals [40].

In view of the conceptual and neural differentiation of social affect and social cognition as well as the findings from (sub-)clinical populations, it is likely that these two subcomponents of social understanding might also be differentially affected by age. Over the last 30 years, a vast body of research emerged on this topic [41]. One of the first studies examining the effect of age on social cognition [42] revealed a preservation or even enhancement of ToM in older compared with younger adults (YA). This finding was followed by an increasing number of studies showing the opposite effect, an age-related decline in taking the perspective of others [43]. However, there are still studies showing a preservation of socio-cognitive processes with age [44–50]. Research on adult age differences in social affect is a little more consistent with most studies showing a preservation or even enhancement in empathy and

compassion [51–54] and only a few a decline [55–57]. Intact or enhanced social affect in older relative to younger adults might be explained by the assumption that emotional goals (e.g. paying attention to emotions) are considered as more relevant and important, and thus are increasingly more selected and pursued as people age [58]. Whereas a decreased social cognition might be explained by the assumption that knowledge-related goals (e.g. pursuit of information about the social world) are considered as less valuable and are selected and pursued less likely which is proposed by the socio-emotional selectivity theory of Carstensen *et al.* [58]. The authors claim that age-related differences in selectively pursuing social goals may arise from altered perception of the remaining available lifetime. Thus, the socio-emotional selectivity theory suggests a shift of motivation from knowledge-related goals to emotional goals from early adulthood to old age in the consequence of a perception of ascendingly constrained time with increasing chronological age [58]. However, most adult development studies until recently focused on the effect of age on *either* the socio-emotional *or* the socio-cognitive route [54,59–61], which precluded the direct comparison of the lifespan development of social affect and cognition. One exception is the study by Reiter *et al.* [62]. The authors investigated age-related differences in empathy, compassion and ToM with an ecological valid paradigm (EmpaToM; [28]) that enables measuring all three abilities within the same person based on the same stimuli. They found that OA showed the same level of empathy and increased compassion compared with YA. In contrast to that, YA outperformed OA in ToM. Thus, Reiter *et al.* [62] observed a preservation or even enhancement for socio-affective processes, but a decline in socio-cognitive processes in old age using the EmpaToM, which is in line with the suggestion of the socio-emotional selectivity theory.

However, with regard to the replication crisis in science in general and in psychological science in particular [63,64], it is important to test the reproducibility of results revealed by each specific method. There is cumulating evidence indicating that about half of the study results in psychology are not replicable [64–66]. Basing our assumptions and knowledge on these results to derive new theories, studies or interventions could have a multitude of negative consequences including misleading predictions, amiss study designs and even ineffective or maladaptive interventions.

The reproducibility of the results of Reiter *et al.* [62] on the effect of age on social affect and cognition would have implications for the development of interventions aiming to improve healthy ageing. A decline in socio-cognitive processes could hinder OA in correctly understanding their interaction partners and thus preclude them from engaging in and successfully managing social interactions. This in turn could cause an overall reduction in social interactions which are so important for mental and physical healthy ageing. Fostering socio-cognitive processes should then become the focus of interventions facilitating social understanding in OA. Therefore, the object of this study was to replicate the effect of age on empathy, compassion and ToM found in the study of Reiter *et al.* [62] by using the same paradigm (EmpaToM) in a different setting (inside an MRI scanner) and in an independent sample of YA and OA but keeping all other methods as close as possible to the original study.

## 2. Methods

This study was part of a larger project consisting of three testing sessions. Here, we focus on the behavioural measures of the third session in which participants performed the EmpaToM task [27,28,67] and another prosocial decision-making task (which will be reported elsewhere) in a 3 tesla MRI scanner as well as a battery of cognitive functioning tasks outside the scanner. The EmpaToM was administered after the prosocial decision-making task. The battery of cognitive functioning tasks was randomly assigned to the participants either before or after the MRI tasks.

### 2.1. Participants

One hundred and one individuals participated in the third session (YA: $N = 45$, age range = 18–30; OA: $N = 56$, age range = 65–78). All participants were recruited via flyers and newspaper announcements in the greater Dresden city region. The OA were additionally recruited from sport, language and university courses as well as choirs for OA and the database of the Lifespan Developmental Neuroscience Lab at Technische Universität Dresden. All participants spoke and understood German fluently, had normal or corrected to normal vision, and were right-handed as well as MRI-suitable individuals (e.g. having no magnetic implants, no self-reported claustrophobia). Additionally, participants were not allowed to have participated in the study of Reiter *et al.* [62] or to have performed the EmpaToM in another study before, to assure an independent sample to Reiter *et al.*'s sample [62]. Because participants in the present study performed the EmpaToM in an MRI scanner, there was a number of additional *a priori* exclusion criteria

**Table 1.** Descriptive and inferential statistics of the sample characteristics.

| | YA | OA | test statistic |
|---|---|---|---|
| years of education | 9.17 (3.70) | 9.26 (4.52) | $t = -0.10384$, $p = 0.9176$ |
| relationship status (partner/single) | 24/17 | 30/11 | $\chi^2 = 1.3558$, $p = 0.2443$ |
| residence (alone/with others) | 12/30 | 12/32 | $\chi^2 = 0$, $p = 1$ |

for the present sample: self-reported psychological or neurological disease currently or within the last 12 month, consumption of more than five cups of coffee a day (in total 1000 ml), smoking of more than five cigarettes per day, drinking of more than 12 g (women)/24 g (men) pure alcohol per day, consumption of illegal drugs more often than two times per month or having a history of drug abuse or addiction, as well as having a prescribed hearing aid device, attested tinnitus or colour blindness. The OA were additionally screened for cognitive impairment (in the first session of the project) via the official German version of the Montreal Cognitive Assessment (MoCA; [68]). With a slight deviation from Reiter *et al.* [62], who used a cut-off of 25, we used a cut-off of 26 on the MoCA [69]. Hence, only OA with a score of 26 or higher on the MoCA were included in the current sample. Fifteen individuals had to be excluded during or after the third session due to an inability to undergo scanning ($N = 5$), problems with hearing or sight during the task ($N = 4$), termination of the session by the participant or due to technical problems ($N = 2$) or neurological abnormalities in the brain ($N = 4$). In contrast to Reiter *et al.* [62], we did not exclude participants ($N = 2$) that performed below chance level (less than 0.33). This did, however, not affect the results in the present study (see electronic supplementary material A for all analyses without these participants).

The final sample comprised 42 younger (21 females, age range = 18–30, $M = 24.00$, s.d. = 3.20) and 44 older (22 females, age range = 65–77, $M = 69.50$, s.d. = 3.74) individuals. Between the two age groups, the gender distribution was comparable. YA and OA did also not differ significantly in their years of education, relationship status or residence (see table 1 for descriptive and inferential statistics) comparable to Reiter *et al.*'s sample [62]. However, this might hint to a slight positive selection bias for the OA in our sample and Reiter *et al.*'s sample [62], since it is not common for OA to have an equivalent level of education as YA in ageing studies. Further, due to the restrictions of an MRI paradigm, our effective sample size was smaller than Reiter *et al.* [62]. Based on the large effects of ageing on socio-affective (compassion) and socio-cognitive (ToM) processes that were found in the study of Reiter *et al.* [62], we conducted a *post hoc* power analysis ($f = 0.40$, two-tailed $\alpha = 0.05$, $n = 86$) which verified that the final sample size was nevertheless appropriate to detect the effects with a power of 1-beta = 0.95.

## 2.2. Materials and procedure

All participants provided written informed consent prior to participation and completed a socio-demographic questionnaire during the first session. In a counterbalanced order, participants then started either with the MRI tasks (including the EmpaToM) and performed the battery of cognitive functioning tasks afterwards, or vice versa. At the end of the third session, they received 8.50€ per hour for participation. The Technische Universität Dresden ethics committee granted ethical approval in accordance with the Helsinki declaration for the whole project (EK 486 112 015).

The EmpaToM [28] is a video-based social interaction task that enables measuring empathy (as the sharing of a narrator's emotion), compassion (as concern for the narrator) and cognitive ToM (via the response to a multiple-choice question about the thoughts of the narrator) within the same person in a single task (figure 1). The EmpaToM starts with a fixation cross (1–3 s) followed by the name of the protagonist (2 s) who will speak about a neutral or emotionally negative life-event in the upcoming short video sequence (approx. 15 s). After each video, participants report how they feel themselves (empathy measure, 7 s) on a visual analogue scale ranging from positive to negative, subsequently followed by the question how much compassion they feel for the protagonist (compassion measure, 7 s) on a visual analogue scale ranging from 'none' to 'very much'. After a short break (a fixation cross, 1–3 s), participants answer a multiple-choice question (max. 25 s) that requires reasoning about the thoughts of the protagonist (ToM measure) or about facts of the story (factual reasoning measure/non-ToM control condition). As in the study of Reiter *et al.* [62], we allowed for an age-adapted response time window of 7 s for the valence and compassion ratings and a maximum of 25 s for the multiple-choice questions. Differing from Reiter *et al.* [62], we did not include a confidence rating after the multiple-choice questions, since we were especially interested in the effect of age on empathy, compassion and ToM,

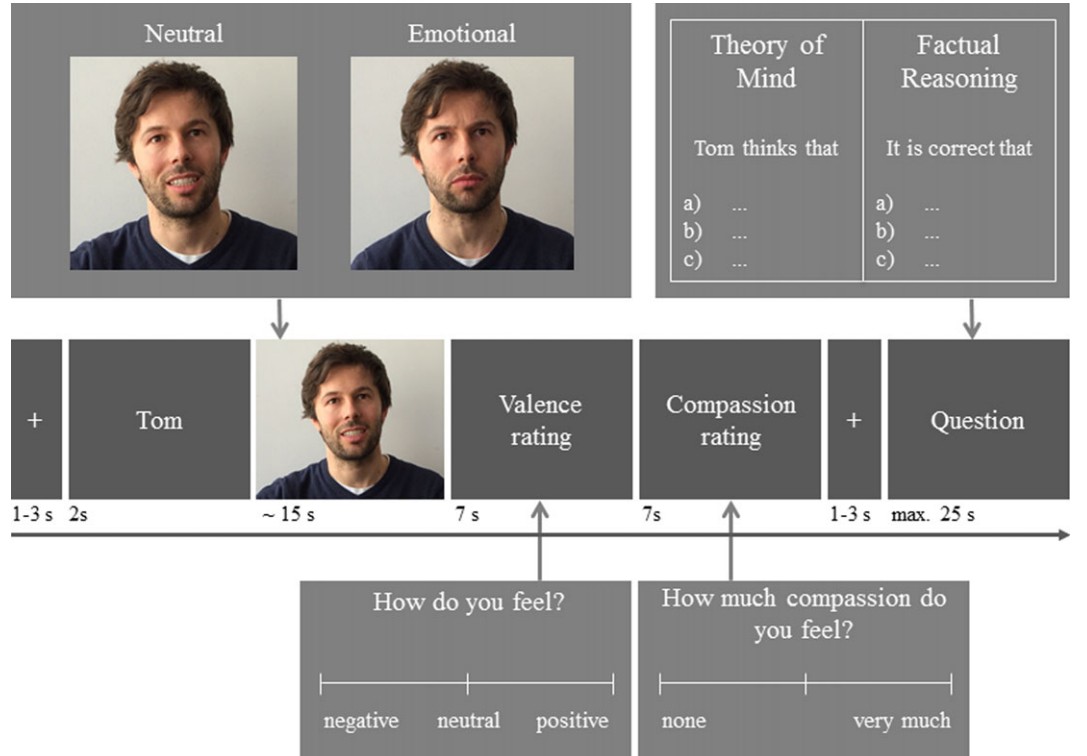

**Figure 1.** EmpaToM. Participants are shown video sequences in which a protagonist speaks about a neutral or emotionally negative life-event. After each video, participants report how they feel themselves and how much they feel for the protagonist. Followed by a multiple-choice question about the thoughts of the protagonist (ToM) or facts of the story (factual reasoning).

rather than metacognition. Prior studies have shown that neither excluding the confidence rating at the end of the trial [40] nor replacing it by another rating [70] affect the previous measures within each trial.

In addition, participants performed a battery of cognitive functioning tasks. These comprised the Trail Making Test A and B (TMT A and B; [71]), the Identical Pictures Test (IDP; [72]), the Digit Span backward (DSb; [73]) and the Spot a Word test (SAW; [74]) to gather attention and complex attention, cognitive speed, working memory and a proxy of verbal intelligence based on lexical decisions, respectively. YA outperformed OA significantly in attention (TMT A) and complex attention (TMT B) as well as in working memory (DSb), coinciding with Reiter et al.'s sample [62]. Regarding the IDP measuring cognitive speed, OA answered significantly slower than YA, but they did not differ significantly in the accuracy of their responses. That OA answered significantly slower than YA on the IDP is in line with Reiter et al.'s sample [62] as well as findings of a population-based lifespan sample [75]. However, that OA answered as accurately as YA on the IDP is not in line with Reiter et al.'s sample [62] and findings of a population-based lifespan sample [75], which again hints to a slight positive selection bias for the OA in our sample. In line with Reiter et al.'s sample [62] and the literature [75,76], OA were as fast as YA and had a higher accuracy than YA on verbal intelligence measures (SAW RT and accuracy). Thus, regarding cognitive abilities the current sample matched the sample of Reiter et al. [62] with comparable means and test statistics on the cognitive measures, except for cognitive speed (IDP accuracy). See table 2 for a summary of these measures.

## 2.3. Data analysis

Age differences in sample characteristics and cognitive measures were examined with independent $t$-tests or $\chi^2$-tests using the R package stats [77]. If the assumptions for an independent $t$-test were violated, a non-parametric test (Mann–Whitney $U$-test) was performed using the R package stats [77]. Regarding the relationship status, four subjects did not report their status or identified with both categories (e.g. married and widowed) and thus were counted as missing value (NA) due to the ambiguity of their response.

Analogous to Reiter et al. [62] measures of empathy (mean valence rating for the emotional and neutral condition separately), compassion (mean concern rating across all conditions) and ToM/

**Table 2.** Descriptive and inference statistics of cognitive measures.

| task | YA mean (s.d.) | OA mean (s.d.). | test statistic | effect size |
|------|------|------|------|------|
| TMT A (time in s) | 23.3 (8.01) | 41.0 (17.8) | $t = -5.975$, $p < 0.001$ | $d = -1.28$ |
| TMT B (time in s) | 42.6 (11.4) | 81.6 (28.6) | $t = -8.377$, $p < 0.001$ | $d = -1.79$ |
| IDP (accuracy) | 0.960 (0.055) | 0.956 (0.053) | $t = 0.281$, $p = 0.780$ | $d = 0.06$ |
| IDP (RT in ms) | 2061 (330) | 3088 (474) | $t = -11.693$, $p < 0.001$ | $d = -2.51$ |
| DSb (total score) | 7.64 (2.20) | 6.25 (1.43) | $t = 3.467$, $p < 0.001$ | $d = 0.751$ |
| SAW (accuracy) | 0.703 (0.076) | 0.804 (0.088) | $t = -5.752$, $p < 0.001$ | $d = -1.24$ |
| SAW (RT in ms) | 3641 (1001) | 3772 (760) | $t = -0.680$, $p = 0.498$ | $d = -0.15$ |

non-ToM (mean reaction time (RT) and error rates (accuracy) for the ToM and non-ToM control condition separately) were derived from the EmpaToM. For the calculation of the mean RT of ToM/non-ToM questions only correct items (excluding false response and no response items) were used. For the calculation of the accuracy of ToM/non-ToM questions no response items were set to zero. Owing to a few technical problems with the button box that had been stuck for a few trials while the EmpaToM was performed, these trials were excluded from all measures, as were sleep trials in which participants did not respond in time on all three measures (valence rating, concern rating, and multiple-choice question). The empathy, compassion and ToM scores were proportionally adjusted for these trials. Furthermore, we excluded three participants (OA = 1, YA = 2) from the analysis of empathy and one participant (YA = 1) from the analysis of compassion due to them being extreme outliers (greater than 3 s.d.) on the valence ratings or concern rating, respectively.

Coherent with Reiter *et al.* [62], the following analyses were conducted. To analyse age differences for empathy, we performed a repeated measures analysis of variance (ANOVA) with emotionality of the video (emotional versus neutral) as within-subject factor and age group (YA versus OA) as between-subject factor (in the following referred to as mixed ANOVA). By analysing the interaction of video type (emotional versus neutral) and age group (YA versus OA) to assess age differences in empathy the effect of general mood is controlled for, since the valence ratings of the neutral videos serve as control conditions that reflect baseline affect. Age differences regarding the capacity of ToM were analysed using a repeated measures ANOVA on accuracy and RT with question type (ToM versus non-ToM) as within-subject factor and age group (YA versus OA) as between-subject factor (in the following referred to as mixed ANOVA). All mixed ANOVAs were performed using the R package rstatix [78]. If the assumption of normality (e.g. for empathy, ToM accuracy) or homogeneity of variances (e.g. ToM RT) was violated, we performed a robust mixed ANOVA using the R package WRS2 [79]. To analyse age differences in compassion, we performed an independent *t*-test using the R package stats [77]. Since the variable compassion had unequal variances in YA and OA, we conducted the non-parametric Mann–Whitney *U*-Test using the R package stats [77] as well.

Also consistent with Reiter *et al.* [62], we further derived a unit-weighted composite score reflecting a proxy of fluid intelligence (based on the z-scores of the TMT A (RT) and B (RT), IDP (mean of RT and accuracy), and DSb (total score)) as well as a unit-weighted composite score reflecting a proxy of verbal intelligence (based on the z-scores of the SAW (RT) and SAW (accuracy)) to test for potential effects of these cognitive abilities on the observed age effects. We performed the repeated measures analysis of covariance (ANCOVA, in the following referred to as two-way ANCOVA) separately for the proxy of fluid intelligence and for the proxy of verbal intelligence with the R package rstatix [78] and the R package WRS2 [79]. All data preparation and analyses were conducted with MATLAB 9.6.0.1214997 R2019a [80] and R 3.6.3 [77].

# 3. Results

## 3.1. Social affect: empathy and compassion

The mixed ANOVA of the valence ratings revealed a significant main effect of emotionality of the video ($F_{1,81} = 675.143$, $p < 0.001$, $\eta p2 = 0.893$), suggesting that emotionally negative videos elicited a stronger empathic response than neutral videos in our participants (figure 2). However, we observed no

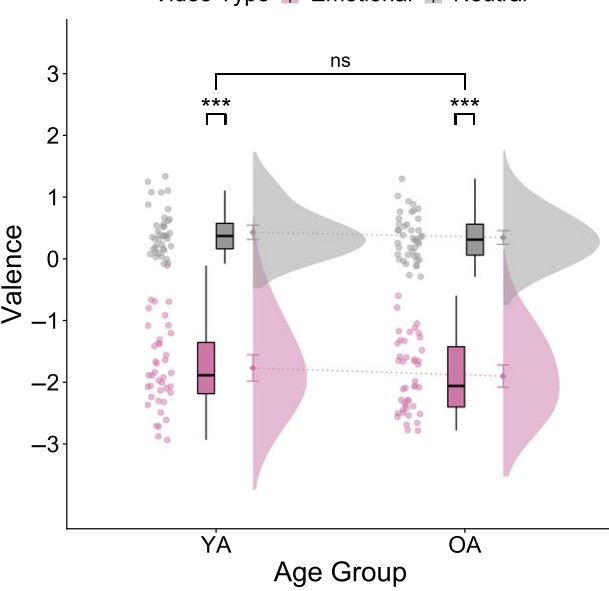

**Figure 2.** Age differences regarding empathy. Valence ratings show a significant effect of video type. YA and OA experience emotionally negative videos as significantly more negative than neutral videos. The effect of age on the valence ratings and the interaction of video type and age group are not significant, indicating that YA and OA show similar empathic responses to a person describing an emotionally negative or neutral situation of their life. Error bars represent 95% confidence intervals.

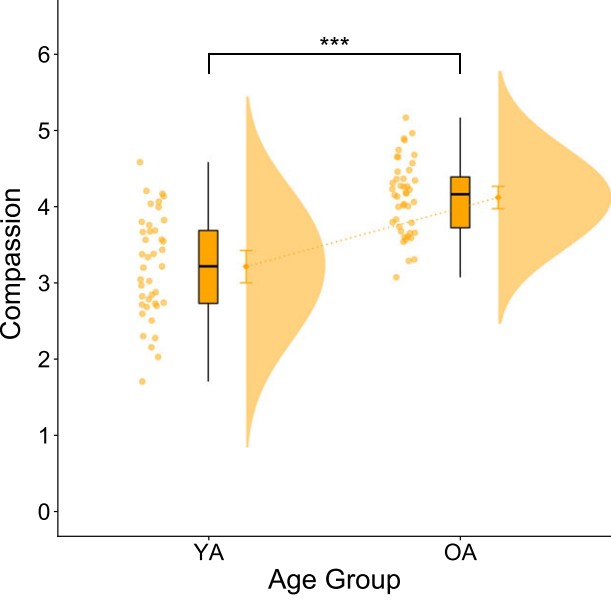

**Figure 3.** Age differences regarding compassion. Compassion ratings show a significant effect of age group. OA report significantly more compassion for the narrators than YA do. Error bars represent 95% confidence intervals.

significant main effect of age group ($F_{1,81} = 2.164$, $p = 0.145$, $\eta p2 = 0.026$) and no significant interaction effect ($F_{1,81} = 0.078$, $p = 0.780$, $\eta p2 = 0.0001$) in our sample. Thus, YA and OA did not differ significantly in their empathic response to emotionally negative and neutral videos. Since the assumption of normality was violated for the valence ratings of the neutral videos in YA and emotional videos in OA, we additionally performed a robust mixed ANOVA that validated the former analysis and replicated the findings of Reiter *et al.* [62].

Regarding the compassion ratings, the independent *t*-test showed a significant age effect ($t_{72} = 7.087$, $p < 0.001$, $d = -1.55$, 95% CI [2.1, 1.15]), with OA reporting more compassion than YA across all conditions (figure 3). Since the assumption of homogeneity of variances was violated for the compassion ratings, we additionally conducted a Mann–Whitney *U*-test that validated the former analysis and replicated the findings of Reiter *et al.* [62].

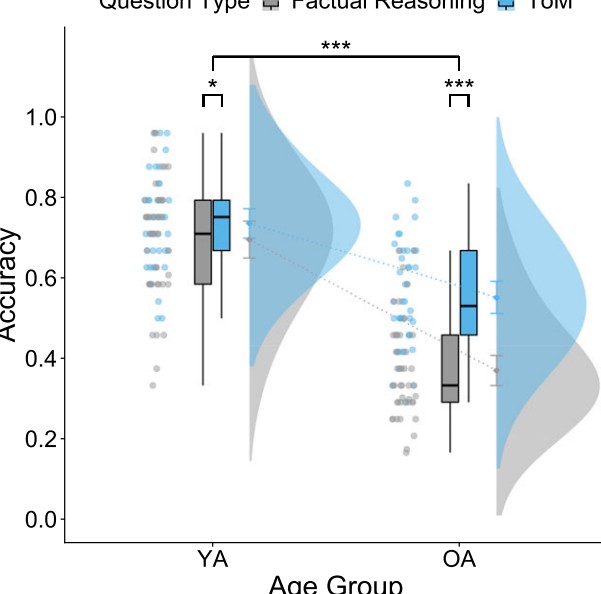

**Figure 4.** Age differences regarding Theory of Mind accuracy. There is a significant main effect of age group for ToM accuracy. YA outperform OA on ToM and factual reasoning (control) questions. A significant main effect of question type shows that YA and OA perform better on ToM question than control questions. A significant age group and question type interaction qualifies the main effects. The difference in performance for ToM versus control questions is significantly stronger in OA compared with YA. Error bars represent 95% confidence intervals.

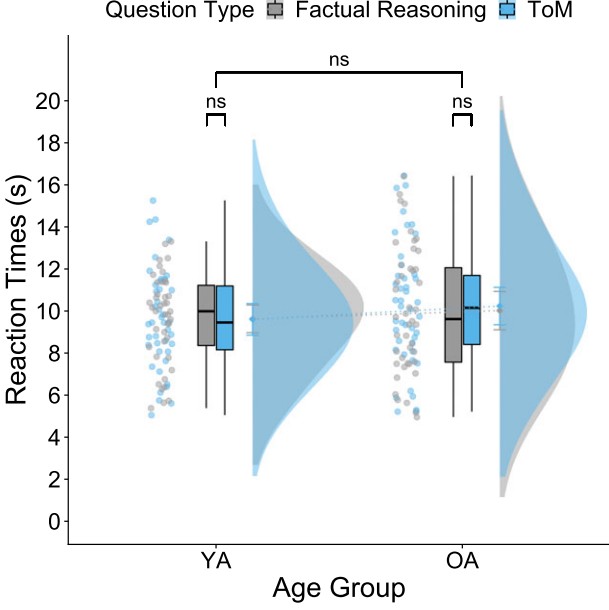

**Figure 5.** Age differences regarding Theory of Mind reaction time. There is no significant main effect of age group or question type nor a significant interaction of both for RTs. YA and OA answered ToM and factual reasoning questions equally fast. Error bars represent 95% confidence intervals.

## 3.2. Social cognition: Theory of Mind

The mixed ANOVA for accuracy revealed a significant main effect of question type ($F_{1,84} = 65.532$, $p < 0.001$, $\eta p2 = 0.438$), with higher accuracies for ToM questions ($t_{85} = 7.20$, $p < 0.001$, $d = 0.777$, 95% CI [−1.01, −0.6]), and a significant main effect of age group ($F_{1,84} = 107.831$, $p < 0.001$, $\eta p2 = 0.562$), with YA answering more accurately than OA ($t_{82} = 10.363$, $p > 0.001$, $d = 2.24$, 95% CI[1.76, 2.91]). We also found a significant interaction between question type and age group ($F_{1,84} = 26.580$, $p < 0.001$, $\eta p2 = 0.240$) that qualified the main effects and is in line with the findings of Reiter *et al.* [62]. Bonferroni-corrected

*post hoc* tests showed that both YA and OA performed significantly better in the ToM condition than in the factual reasoning condition, but the effect was stronger for OA ($t_{43} = -8.95$, $p = 0.001$, $d = -1.43$, 95% CI [−1.99, −1.01]) than YA ($t_{41} = -2.20$, $p = 0.034$, $d = -0.302$, 95% CI [−0.76, 0.14]) (figure 4). Thus, the decrease in performance for the factual reasoning condition compared with the ToM condition was significantly stronger for OA. This finding differs slightly from the finding of Reiter *et al.* [62], who found no significant difference in accuracy for the ToM and factual reasoning condition in YA. Since the assumption of normality was violated for the factual reasoning condition in OA, we additionally conducted a robust mixed ANOVA for accuracy that validated our former analysis.

The mixed ANOVA for RTs showed neither a significant main effect of question type ($F_{1,84} = 0.570$, $p = 0.452$, $\eta p2 = 0.007$) nor of age group ($F_{1,84} = 0.842$, $p = 0.361$, $\eta p2 = 0.010$), suggesting that ToM and factual reasoning questions were answered similarly fast by YA and OA in our sample and replicates Reiter *et al.*'s results [62]. The interaction between question type and age group was not significant either ($F_{1,84} = 0.906$, $p = 0.344$, $\eta p2 = 0.011$) (figure 5). This finding differs slightly from Reiter *et al.*'s results [62], who found a significant question type x age group interaction in RT. Since the assumption of homogeneity of variances was violated for the factual reasoning conditions, we additionally performed a robust mixed ANOVA for RT. This robust analysis showed a slightly significant interaction between question type and age group ($F_{1,44} = 4.2743$, $p = 0.0446$); however, robust *post hoc* tests on the difference scores (ToM–factual reasoning) did not reveal a significant difference in RT as a function of age group (estimate = 0.2952, $p = 0.208$).

## 3.3. Analysis of covariates

To test for potential effects of fluid intelligence and verbal intelligence on the observed age effects on ToM/factual reasoning accuracy and compassion as well as the absence of an age effect on empathy, we conducted separate mixed ANCOVAs, with either the proxy of fluid intelligence or the proxy of verbal intelligence as covariate. Controlling for these covariates separately, yielded the same age effects as observed above on ToM/factual reasoning accuracy (all $Fs_{1,83} \geq 18.123$, all $ps < 0.001$, all $\eta p2 \geq 0.179$) and compassion (all $Fs_{1,82} \geq 19.066$, all $ps < 0.001$, all $\eta p2 \geq 0.189$). The separate adjustment for the proxy of fluid intelligence or the proxy of verbal intelligence did not alter the absence of an age effect on empathy either (video type: all $Fs_{1,80} = 667.949$, $p < 0.001$, $\eta p2 \geq 0.893$; all other $Fs_{1,80} \leq 1.247$, all other $ps \geq 0.267$, all other $\eta p2 \leq 0.015$). Since the assumption of homogeneity of variances was violated for empathy as well as for compassion, we additionally conducted robust ANCOVAs that validated our former analysis. Thus, adding a proxy of fluid intelligence or verbal intelligence as covariate in the analyses did not alter the effects of age on ToM/factual reasoning accuracy and compassion or the absence of an effect of age on ToM/factual RT and empathy, which is in line with Reiter *et al.* [62].

# 4. Discussion

In this study, we aimed to replicate the effects of age on empathy, compassion and ToM observed in the study of Reiter *et al.* [62] by using the same paradigm (EmpaToM) in an independent sample of YA and OA. For both components of social understanding that are measured by the EmpaToM, namely social affect and social cognition, we mainly succeeded in replicating the results.

## 4.1. Social affect

We assessed socio-affective processes after the participants had watched a short video of a protagonist talking about an emotionally negative or neutral life-event by measuring empathy (as the sharing of a narrator's emotion) and compassion (as concern for the narrator). For empathy we found a significant main effect of video type on the valence ratings, showing that emotionally negative videos are experienced as more negative than neutral videos in both YA and OA (which is consistent with the general effect of video type of the task and shows that the videos are observed as emotionally different; [28]). The main effect of age on the valence ratings and the interaction of video type and age were not significant. The absence of a significant interaction effect may be interpreted as YA and OA showing similar empathic responses to a person describing an emotionally negative or neutral situation of their life. Moreover, for compassion we found a significant effect of age, showing that OA report significantly more compassion for the narrators than YA do. These results are in line with Reiter *et al.* [62], indicating a preservation or even enhancement in socio-affective processes with age.

## 4.2. Social cognition

We assessed socio-cognitive processes after the participants had watched a short video of a protagonist talking about an emotionally negative or neutral life-event by measuring ToM (as the reasoning about a narrator's thought). Factual reasoning (as the reasoning about facts included in the story) was used as control condition. Regarding the accuracy for ToM and factual reasoning, we found a significant main effect of age, revealing that YA outperform OA in both conditions. We also found a significant main effect of question type, showing that both YA and OA perform better for ToM versus factual reasoning questions. The main effects as well as a significant interaction of question type and age that qualified the main effects are in line with Reiter *et al.*'s findings [62]. *Post hoc t*-tests on the interaction effect differed slightly from Reiter *et al.*'s results [62]. OA performed significantly better in the ToM versus the factual reasoning condition in both the current sample as well as in Reiter *et al.*'s sample [62]. However, in contrast to Reiter *et al.* [62], in the current study, YA differed in accuracy as a function of question type (factual reasoning versus ToM). YA in our sample performed significantly better in the ToM versus the factual reasoning condition. Yet, slight differences in accuracy between independent samples of YA have already been reported by Kanske *et al.* [28]. Nevertheless, the difference in performance for ToM versus factual reasoning questions is significantly larger in OA compared with YA leading to the observed interaction effect of age and question type. This is caused by a remarkable decline in accuracy for factual reasoning questions in OA. Despite the slight discrepancies within the younger age group, we were able to replicate the results in the OA group, which suggests that ToM is less impaired by ageing than factual reasoning. This points to an advantage of social as compared with factual cognition in OA.

Regarding the response time for the ToM and factual reasoning questions, we found neither a significant main effect of age nor of question type, revealing that both age groups respond equally fast, and both question types are answered equally quickly. This is again in line with Reiter *et al.*'s results [62]. However, we did not find a statistically significant interaction of question type and age group that Reiter *et al.* [62] had reported. In the study of Reiter *et al.* [62], OA performed equally fast on ToM questions as YA, but OA responded significantly slower on factual reasoning questions. One could argue that the difference between the two studies regarding the interaction effect of reaction times was due to the usage of different keyboards. Reiter *et al.* [62] had participants sit in front of a computer answering the questions on a regular keyboard, whereas our participants lay in an MRI scanner answering the questions on an MRI-suitable keyboard that rested on their right thigh with their fingers directly on the keys. However, it is unlikely that the usage of different keyboards caused such a specific task effect. It seems more likely that the effect of age on response time is not as stable and reliable as for accuracy measures. This could be due to the long response time windows of the task (25 s) in the current study and the study by Reiter *et al.* [62], which may have caused participants to focus on correct rather than fast responses, thereby decreasing the information content of the response time.

## 4.3. Analysis of covariates

To control for potential effects of diverging cognitive abilities between the age groups, we conducted separate ANCOVA analyses for empathy, compassion and ToM with either a proxy of fluid intelligence or a proxy of verbal intelligence as covariate. These control analyses did not qualitatively change the observed results which are in line with Reiter *et al.*'s results [62]. This indicates that neither fluid intelligence nor verbal intelligence affect the age-related changes in empathy, compassion or ToM, at least if they are measured with the tasks we used in current study. However, various studies point to the importance of inhibitory control for empathy and ToM in YA (for a meta-analysis see [81]) as well as for the effect of age on ToM (e.g. [59,82]). As a cognitive process that enables us to inhibit or control our attention, thoughts, emotions and behaviour such as egocentricity, inhibitory control seems to be essential to share the feeling of others as well as to take their perspectives [83,84]. The question, whether the effect of ageing on socio-affective and socio-cognitive processes is not affected by the ageing of executive function processes, should, therefore, be addressed in future research using comprehensive measures of these cognitive abilities.

## 4.4. Taken together

We were able to replicate the effect of age on empathy and compassion reported by Reiter *et al.* [62]—revealing a preservation of empathy and enhancement of compassion in OA. Concerning ToM we

replicated the effect of age on ToM accuracy—showing a decline in ToM performance in OA compared with YA. Age differences with respect of ToM are of notable interest given comparable educational attainment levels in YA and OA in the current sample (coherent with Reiter *et al*.'s sample [62]). This speaks against the notion proposed by several authors [85,86] that age differences in perspective taking are mediated through differing educational levels in these cohorts. However, we did not replicate the effect of age on ToM reaction time.

Nevertheless, the current study has three main limitations that would be important to address in future research. First, testing high-functioning individuals in our sample with equal levels of education in YA and OA, can be regarded as both a strength and a limitation. The latter particularly relates to limitations with respect to external validity or the question whether the findings can be generalized towards other populations. In a similar vein, we note that the current sample is, as in most previous research, a 'western, educated, industrial, rich and democratic' (WEIRD) sample [86], which is not comparable to the total global population and hence prevents a generalization of the results, since cultural differences have great impact on our development [88] and there is evidence that empathy and ToM development and manifestation vary across cultures [89–97]. Thus, we should, when referring to the results of this study and most other studies, probably speak about 'the ageing of the WEIRD social mind'. Second, with the current study and the study of Reiter *et al*. [62], as well as most studies on this topic we cannot rule out that the results for empathy, compassion and ToM are caused by cohort effects. This limitation could only be addressed by longitudinal studies. As a longitudinal study across the whole adult lifespan takes longer than most researchers' careers, that is arguably an ambitious aim that should nevertheless be taken seriously. Third, the current study and the study of Reiter *et al*. [62] as well as most studies on this topic included only YA and OA, but not the entire adult age range. This limitation might skew the whole picture of lifespan development of empathy, compassion and ToM. It may preclude insights into the shape and pace of the developmental trajectory of social development—i.e. are the gains in compassion, or the decrease of ToM linear over the lifespan? Studies that already investigated perspective taking across a wide range of the adult lifespan showed that YA typically outperform OA, yet middle-age adults outperform both former groups [98,99]. Thus, future research should focus on samples covering the whole adult lifespan rather than comparing only two age groups to reveal if the effect of age on ToM equals a linear decline or an inverted-U-shape. The same should be considered for empathy and compassion to answer the question if empathic capacity is the same across adulthood and if compassion increases continuously. A combination of both—longitudinal studies that measure socio-affective and socio-cognitive processes at least over one decade of adulthood and cross-sectional samples across the entire adult lifespan and across diverse cultures—would be the most sophisticated approaches targeting these limitations.

Despite these limitations the current study lends more evidence to the view that socio-affective and socio-cognitive processes age differentially across the adult lifespan. Further, in view of the replication crisis in psychological science, reproducing the results of a study with an independent sample in a different setting (e.g. here MRI versus test cabins) has great value not only for science itself, but also for the reliability of the results and thus new ideas, theories or interventions that can be derived from these results. Overall, our findings support the assumption that the socio-affective processes are preserved or even enhanced while socio-cognitive processes decline with age, which is derived from the socio-emotional selective theory [58] in the ageing literature. This may have implications for future research that aims to promote healthy ageing by facilitating successful social relationships. Considering that the capacity to infer the mental states of others seems to be decreased in OA and that this socio-cognitive process seems to be essential for adaptive social interactions [100–103], studies might use and transfer ToM trainings [104,105] to improve perspective taking in high age individuals or even prevent the decline in this socio-cognitive process.

Ethics. The Technische Universität Dresden ethics committee granted ethical approval in accordance with the Helsinki declaration for the whole project (EK 486 112 015) and the research was conducted according to the principles expressed in the declaration of Helsinki. All participants provided written informed consent prior to participation.

Data accessibility. This article received results-blind in-principle acceptance (IPA) at *Royal Society Open Science*. Following IPA, the accepted Stage 1 version of the manuscript, not including results and discussion, was preregistered on the Open Science Framework (https://doi.org/10.17605/OSF.IO/9BF3S). This preregistration was performed after data analysis. All data, code and materials associated with this project are also available on the Open Science Framework (https://doi.org/10.17605/OSF.IO/7EDBN).

The data are provided in electronic supplementary material [106].

Authors' contributions. A.M.F.R., S.-C.L. and P.K. developed the study concept, P.K. developed the task, and J.S., M.K., A.M.F.R., S.C.L. and P.K. contributed to the study design. J.S., L.P. and M.K. collected the data, J.S. and L.P.

conducted the data preparation, and J.S analysed the data. J.S., L.P., A.M.F.R. and P.K. interpreted the data, J.S. drafted the manuscript, and L.P., S.-C.L., A.M.F.R. and P.K. provided critical revisions. All authors approved the final version of the manuscript.

Competing interests. The authors declare that the research was conducted in the absence of any commercial or financial relationships that could be construed as a potential conflict of interest.

Funding. This work was supported by a grant from the German Research Foundation (DFG) awarded to P.K., S.-C.L., and A.M.F.R. (DFG Project ID: 427083324). Open Access Funding by the Publication Fund of the TU Dresden.

Acknowledgments. We thank M. Schmitz, S. Valdivieso, I. Zimmermann, J. Stein, M. Hildebrandt, S. Geißler, S. Giese, R. Hamburg, C. Margraf, R. Krug, K. Raum and V. Wetzlich for their assistance with data acquisition and all participants for study participation.

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
