## [Peer Review File · Royal Society Open Science]

Review History

RSOS-202300.R0 (Original submission)

Review form: Reviewer 1

Do you have any ethical concerns with this paper?

No

Have you any concerns about statistical analyses in this paper?

No

Recommendation?

Major revision

Comments to the Author(s)

This study aims to replicate the findings of Reiter et al. (2017) who showed that socio-affective processes are preserved in older age while socio-cognitive processes decline.

The Introduction is well written and the Method is described in sufficient detail.

I was surprised to see that the authors are not including confidence ratings (i.e., to index social metacognition) in this replication study. This is a limitation, given that, as Reiter et al. explain “social metacognition as an important facet of social information processing has yet to be studied from an adult developmental perspective.”. To my knowledge, Reiter et al. (2017) is still the only study to have looked at social metacognition in aging to date, and they found that metacognition age deficits were evident in factual but not social reasoning – this is an important and interesting finding that needs to be replicated. The inclusion of the confidence/metacognition ratings in the current study would provide a much stronger contribution to the literature.

The authors define ‘empathy’ as ‘feeling with somebody’. This is correct for affective empathy but not cognitive empathy. I think it would be helpful to explain that it is generally agreed that empathy is a multifaceted construct that has cognitive and affective components, and that in this study empathy is referring exclusively to the affective component of empathy. It might also be useful to mention that cognitive empathy overlaps with theory of mind.

Review form: Reviewer 2

Do you have any ethical concerns with this paper?

No

Have you any concerns about statistical analyses in this paper?

No

Recommendation?

Accept in principle

Comments to the Author(s)

The present study is a replication of a previous research investigating the age-related differences in socio-affective and socio-cognitive processes. I found the topic very interesting and the focus of the investigation particularly relevant within the current effort to study affective and cognitive processes in aging.

The paper is well written. My only concern regards the use of the word "naturalistic" as definition of the EmpaToM task. I'd suggest to use ecological task instead of naturalistic task.

Review form: Reviewer 3

Do you have any ethical concerns with this paper?

Yes

Have you any concerns about statistical analyses in this paper?

No

Recommendation?

Reject

Comments to the Author(s)

Introduction: For a study examining social cognition and ageing, the authors severely underrepresent the vast and interesting literature on this topic. A reader might think this is an under researched area, but it is not, and this needs to be addressed.

The socio-emotional selective theory by Carstensen et al. is considered a theory of motivation and attention in ageing, to my knowledge it does not state 'that cognitive components decline with age but emotional components are preserved or even enhance with age' as the authors argue in their introduction. I would recommend some clarification here and carefully consider the relationship between social cognition and attentional/motivation theories of ageing as they are quite different.

Review form: Reviewer 4

Do you have any ethical concerns with this paper?

No

Have you any concerns about statistical analyses in this paper?

No

Recommendation?

Accept with minor revision

Comments to the Author(s)

Social cognition involves the social cognitive processes that allow individuals to interpret the behaviors of others. These processes include socio-affective processes such as empathy and compassion and socio-cognitive processes such as theory of mind (ToM). A previous study by Reiter et al. has shown that older adults show similar empathy and increased compassion for other people compared to younger adults but have significantly poorer theory of mind (ToM). It is important to replicate these findings as they have implications for interventions designed to aid healthy aging. In the current study, the authors attempt to replicate the Reiter et al. findings by administering the EmpaToM Task (as in the original study) to healthy younger and older adults but inside the MRI scanner, as part of a larger scale study.

This is an interesting and important replication. However, I have some initial comments about the study:

- The authors discuss theory of mind (ToM) as one component of social cognition. However, as we know, ToM can be subdivided into affective and cognitive ToM. It is a little unclear how this relates to the socio-affective and socio-cognitive processes they discuss in their paper. I think it's important to explain this within the study and be clear how it relates to the processes they are investigating, especially as age effects are consistently found in the aging literature on cognitive ToM but less consistently so for affective ToM. I realize Reiter et al. did not do this but I think it would improve how these findings relate to the social cognition literature.
- It is also not clear from the EmpaToM Task which aspect of ToM it is assessing. From the methods, it appears to be examining cognitive rather than affective ToM but this needs some clarity. I did wonder whether the empathy measure was assessing affective ToM but the questions suggest that participants indicated how they felt rather than the speaker in the video.
- The participants in the current replication were studied inside the scanner. It is not clear to me why this was necessary and I do have concerns about the change in the testing environment compared to Reiter et al. This change in the environment might have an influence

on older adults' performance as the scanner could be conceived as a stressful, unusual environment. Were these participants used to being inside a scanner? If you fail to replicate, might this be why?

- Having an equivalent level of education in the younger and older groups is certainly unusual. Why might this have occurred? How did the education levels compare with Reiter et al.? This is important because studies have shown that higher education protects against the impact of healthy and pathological aging. Similarly, it's a little unusual for older adults not to be impaired in terms of speed. If the authors fail to replicate, might it be something related to their slight positive selection bias?
- Are there issues with the EmpaToM Task having self-report measures for the empathy and compassion questions but an objective measure of ToM?
- What happened if participants didn't respond within the time limits? Was that item excluded and the score proportionally adjusted or did participants receive a score of zero?
- It is unclear why fluid and verbal measures were considered in separate ANCOVAs. I may have misunderstood but I think Reiter et al considered them in the same analyses?
- Were any mood measures included as this might influence performance? I know that Reiter administered the PANAS.

Decision letter (RSOS-202300.R0)

Dear Ms Stietz,

The Editors assigned to your Stage 1 Replication submission ("The Aging of the Social Mind: Replicating the preservation of socio-affective and the decline of socio-cognitive processes in old age") have now received comments from reviewers. We would like you to revise your paper in accordance with the referee and editors suggestions which can be found below (not including confidential reports to the Editor). Please note this decision does not guarantee eventual acceptance.

Please submit a copy of your revised paper within three weeks (i.e. by the 02-Mar-2021). If deemed necessary by the Editors, your manuscript will be sent back to one or more of the original reviewers for assessment.

When submitting your revised manuscript, you must respond to the comments made by the referees and upload a file "Response to Referees" in the "File Upload" step. Please use this to document how you have responded to the comments, and the adjustments you have made. In order to expedite the processing of the revised manuscript, please be as specific as possible in your response.

Once again, thank you for submitting your manuscript to Royal Society Open Science and I look forward to receiving your revision. If you have any questions at all, please do not hesitate to get

in touch. Full author guidelines may be found at <https://royalsocietypublishing.org/rsos/replication-studies#AuthorsGuidance>.

Kind regards,
 Professor Chris Chambers
 Royal Society Open Science
openscience@royalsociety.org

on behalf of Professor Chris Chambers (Registered Reports Editor, Royal Society Open Science)
openscience@royalsociety.org

Associate Editor Comments to Author (Professor Chris Chambers):

Associate Editor: 1

Comments to the Author:

Four expert reviewers have now assessed the Stage 1 manuscript. All reviewers broadly judge Stage 1 primary criterion #1 ("Whether the authors provide a sufficiently clear and detailed description of the methods") to met met, although there are still areas where adherence to this criteria can be improved (see comments of Reviewer 4, especially). As you will see, the reviews are more critical of Stage 1 primary criterion #2 ("Whether the manuscript describes a sufficiently valid (i.e. close) and robust (e.g. statistically powerful) replication of the original study methods and rationale to provide an indication of replicability").

The reviewers also request clarifications to the theoretical framing (Reviewer 3), terminology (Reviewer 2) and consideration of the validity of the design without additional measurements (Reviewer 1). In revising please note that unlike regular articles (or even Registered Reports), for the Replications article type at Royal Society Open Science, the key focus is on ensuring as close as possible a match between the methods of the original study and the replication, even when the methods of the original study may be suboptimal. Limitations in the design can then be addressed at Stage 2 in various ways (see <https://royalsocietypublishing.org/rsos/replication-studies> for details).

Comments to Author:

Reviewer: 1

Comments to the Author(s)

This study aims to replicate the findings of Reiter et al. (2017) who showed that socio-affective processes are preserved in older age while socio-cognitive processes decline. The Introduction is well written and the Method is described in sufficient detail.

I was surprised to see that the authors are not including confidence ratings (i.e., to index social metacognition) in this replication study. This is a limitation, given that, as Reiter et al. explain "social metacognition as an important facet of social information processing has yet to be studied from an adult developmental perspective.". To my knowledge, Reiter et al. (2017) is still the only study to have looked at social metacognition in aging to date, and they found that metacognition age deficits were evident in factual but not social reasoning – this is an important and interesting finding that needs to be replicated. The inclusion of the confidence/metacognition ratings in the current study would provide a much stronger contribution to the literature.

The authors define 'empathy' as 'feeling with somebody'. This is correct for affective empathy but not cognitive empathy. I think it would be helpful to explain that it is generally agreed that empathy is a multifaceted construct that has cognitive and affective components, and that in this study empathy is referring exclusively to the affective component of empathy. It might also be useful to mention that cognitive empathy overlaps with theory of mind.

Reviewer: 2

Comments to the Author(s)

The present study is a replication of a previous research investigating the age-related differences in socio-affective and socio-cognitive processes. I found the topic very interesting and the focus of the investigation particularly relevant within the current effort to study affective and cognitive processes in aging.

The paper is well written. My only concern regards the use of the word "naturalistic" as definition of the EmpaToM task. I'd suggest to use ecological task instead of naturalistic task.

Reviewer: 3

Comments to the Author(s)

Introduction: For a study examining social cognition and ageing, the authors severely underrepresent the vast and interesting literature on this topic. A reader might think this is an under researched area, but it is not, and this needs to be addressed.

The socio-emotional selective theory by Carstensen et al. is considered a theory of motivation and attention in ageing, to my knowledge it does not state 'that cognitive components decline with age but emotional components are preserved or even enhance with age' as the authors argue in their introduction. I would recommend some clarification here and carefully consider the relationship between social cognition and attentional/motivation theories of ageing as they are quite different.

Reviewer: 4

Comments to the Author(s)

Social cognition involves the social cognitive processes that allow individuals to interpret the behaviors of others. These processes include socio-affective processes such as empathy and compassion and socio-cognitive processes such as theory of mind (ToM). A previous study by Reiter et al. has shown that older adults show similar empathy and increased compassion for other people compared to younger adults but have significantly poorer theory of mind (ToM). It is important to replicate these findings as they have implications for interventions designed to aid healthy aging. In the current study, the authors attempt to replicate the Reiter et al. findings by administering the EmpaToM Task (as in the original study) to healthy younger and older adults but inside the MRI scanner, as part of a larger scale study.

This is an interesting and important replication. However, I have some initial comments about the study:

- The authors discuss theory of mind (ToM) as one component of social cognition. However, as we know, ToM can be subdivided into affective and cognitive ToM. It is a little unclear how this relates to the socio-affective and socio-cognitive processes they discuss in their paper. I think it's important to explain this within the study and be clear how it relates to the processes they are investigating, especially as age effects are consistently found in the aging literature on cognitive ToM but less consistently so for affective ToM. I realize Reiter et al. did not do this but I think it would improve how these findings relate to the social cognition literature.
- It is also not clear from the EmpaToM Task which aspect of ToM it is assessing. From the methods, it appears to be examining cognitive rather than affective ToM but this needs some clarity. I did wonder whether the empathy measure was assessing affective ToM but the questions suggest that participants indicated how they felt rather than the speaker in the video.
- The participants in the current replication were studied inside the scanner. It is not clear to me why this was necessary and I do have concerns about the change in the testing environment compared to Reiter et al. This change in the environment might have an influence on older adults'

performance as the scanner could be conceived as a stressful, unusual environment. Were these participants used to being inside a scanner? If you fail to replicate, might this be why?

- Having an equivalent level of education in the younger and older groups is certainly unusual. Why might this have occurred? How did the education levels compare with Reiter et al.? This is important because studies have shown that higher education protects against the impact of healthy and pathological aging. Similarly, it's a little unusual for older adults not to be impaired in terms of speed. If the authors fail to replicate, might it be something related to their slight positive selection bias?
- Are there issues with the EmpaToM Task having self-report measures for the empathy and compassion questions but an objective measure of ToM?
- What happened if participants didn't respond within the time limits? Was that item excluded and the score proportionally adjusted or did participants receive a score of zero?
- It is unclear why fluid and verbal measures were considered in separate ANCOVAs. I may have misunderstood but I think Reiter et al considered them in the same analyses?
- Were any mood measures included as this might influence performance? I know that Reiter administered the PANAS.

Author's Response to Decision Letter for (RSOS-202300.R0)

See Appendix A.

RSOS-202300.R1 (Revision)

Review form: Reviewer 1

Do you have any ethical concerns with this paper?

No

Have you any concerns about statistical analyses in this paper?

No

Recommendation?

Reject

Comments to the Author(s)

I would like to thank the authors for responding to my concerns.

I appreciate that the omission of the confidence ratings does not impact the other measures in the task (e.g., ToM, empathy). However, if the goal is to replicate the methods of the original study by Reiter et al. (2017) as closely as possible, this study has not achieved that with the current methodology. Meta-cognition is a key focus of the original study (discussed throughout the whole manuscript by Reiter et al., including the abstract) but it has been completely omitted here. Without the inclusion of this data, this study appears to only be a partial replication of Reiter et al. (2017). Perhaps this should be made more clear in the Introduction.

Minor point:

I noticed there is a small difference in the reporting of the MoCA screening cut-off scores that the authors might want to clarify. The authors state: "Analogous to Reiter and colleagues, only OA with a score of 26 or higher were included in the current sample" but Reiter et al. reported that participants who scored below 25 on the MoCA were excluded. This would mean that people who scored 25 and above on the MoCA would have been included in Reiter et al.? I think it is fine if there is a difference but perhaps the authors should change their wording to make it clear that this was not the same cut off used in the original paper by Reiter et al.

Review form: Reviewer 4

Do you have any ethical concerns with this paper?

No

Have you any concerns about statistical analyses in this paper?

No

Recommendation?

Accept in principle

Comments to the Author(s)

I am happy that the authors have addressed the comments of the reviewers.

Decision letter (RSOS-202300.R1)

Dear Ms Stietz

On behalf of the Editors, I am pleased to inform you that your Manuscript RSOS-202300.R1 entitled "The Aging of the Social Mind: Replicating the preservation of socio-affective and the decline of socio-cognitive processes in old age" deemed suitable for in-principle acceptance in Royal Society Open Science subject to minor revision in accordance with the referee and editor suggestions. Please find their comments at the end of this email.

The reviewers and handling editors have recommended publication, but also suggest some minor revisions to your manuscript. Therefore, I invite you to respond to the comments and revise your manuscript.

Please you submit the revised version of your manuscript within 7 days (i.e. by the 14-Apr-2021). If you do not think you will be able to meet this date please let me know immediately.

Full author guidelines can be found here <https://royalsocietypublishing.org/rsos/replication-studies#AuthorsGuidance>

on behalf of Professor Chris Chambers
(Subject Editor, Royal Society Open Science)
openscience@royalsociety.org

Associate Editor Comments to Author (Professor Chris Chambers):

Two of the four reviewers who assessed the first submission kindly returned to assess the revised Stage 1 manuscript. As you will see, Reviewer 4 is now satisfied and recommends IPA, whereas Reviewer 1 remains concerned about the omission of confidence ratings, and on this basis judges that Stage 1 primary criterion #2 is unmet (including especially the methodological proximity of the replication to the original study).

Meeting the primary criteria is important for Stage 1 Replications, and so I have looked closely at the reviewer's concern, the revised manuscript, and the authors' previous response to this point. On balance, while I do share the reviewer's concern, I think that even in spite of this omission, the replication is sufficiently close to the original study for the research to be partially informative about replicability, and I have therefore decided not to reject the manuscript on this basis. However, as the reviewer suggests, I would like the authors to be more explicit about the partial nature of the replication.

In revising, please also address the other concern raised by Reviewer 1 concerning the inclusion criteria.

Provided the authors are able to address the above points in a final Stage 1 revision, in-principle acceptance should be forthcoming without requiring further in-depth Stage 1 review.

Reviewer comments to Author:
Reviewer: 1
Comments to the Author(s)

I would like to thank the authors for responding to my concerns.

I appreciate that the omission of the confidence ratings does not impact the other measures in the task (e.g., ToM, empathy). However, if the goal is to replicate the methods of the original study by Reiter et al. (2017) as closely as possible, this study has not achieved that with the current methodology. Meta-cognition is a key focus of the original study (discussed throughout the whole manuscript by Reiter et al., including the abstract) but it has been completely omitted here. Without the inclusion of this data, this study appears to only be a partial replication of Reiter et al. (2017). Perhaps this should be made more clear in the Introduction.

Minor point:

I noticed there is a small difference in the reporting of the MoCA screening cut-off scores that the authors might want to clarify. The authors state: "Analogous to Reiter and colleagues, only OA with a score of 26 or higher were included in the current sample" but Reiter et al. reported that participants who scored below 25 on the MoCA were excluded. This would mean that people who scored 25 and above on the MoCA would have been included in Reiter et al.? I think it is fine if there is a difference but perhaps the authors should change their wording to make it clear that this was not the same cut off used in the original paper by Reiter et al.

Reviewer: 4

Comments to the Author(s)

I am happy that the authors have addressed the comments of the reviewers.

Author's Response to Decision Letter for (RSOS-202300.R1)

See Appendix B.

Decision letter (RSOS-210641.R0)

Dear Mrs Stietz

On behalf of the Editor, I am pleased to inform you that your Manuscript RSOS-210641 entitled "The Aging of the Social Mind: Replicating the preservation of socio-affective and the decline of socio-cognitive processes in old age" has been accepted in principle for publication in Royal Society Open Science.

Please note that you must now register your approved protocol on the Open Science Framework (<https://osf.io/rr>), using the 'Submit your approved Registered Report' option and then the 'Registered Report Protocol Preregistration' option. Please use the Registered Report option even though your article is being accepted as a Stage 1 Replication. Further into the registration process, in the Journal Title field enter 'Royal Society Open Science (Replication article type, Results-Blind track)'. Please note that a time-stamped, independent registration of the protocol is mandatory under journal policy, and manuscripts that do not conform to this requirement cannot be considered at Stage 2. The protocol should be registered unchanged from its current approved state. Please include a URL to the protocol in your Stage 2 manuscript, and because you

submitted via the Results-Blind track please note in the manuscript that the pre-registration was performed after data analysis (e.g. 'This article received results-blind in-principle acceptance (IPA) at Royal Society Open Science. Following IPA, the accepted Stage 1 version of the manuscript, not including results and discussion, was preregistered on the OSF (URL). This preregistration was performed after data analysis.')

Following completion of your study, we invite you to resubmit your paper for peer review as a Stage 2 Replication. Please note that your manuscript can still be rejected for publication at Stage 2 if the Editors consider any of the following conditions to be met:

- The Introduction and methods deviated from the approved Stage 1 submission (required).
- The authors' conclusions were not considered justified given the data.

We encourage you to read the complete guidelines for authors concerning Stage 2 submissions at : <https://royalsocietypublishing.org/rsos/replication-studies#AuthorsGuidance>. Please especially note the requirements for data sharing and that withdrawing your manuscript will result in publication of a Withdrawn Registration.

We encourage you to read the complete guidelines for authors concerning Stage 2 submissions at <https://royalsocietypublishing.org/rsos/registered-reports#ReviewerGuideRegRep>. Please especially note the requirements for data sharing and that withdrawing your manuscript will result in publication of a Withdrawn Registration.

Once again, thank you for submitting your manuscript to Royal Society Open Science and I look forward to receiving your Stage 2 submission. If you have any questions at all, please do not hesitate to get in touch. We look forward to hearing from you shortly with the anticipated submission date for your stage two manuscript.

Kind regards,
Professor Chris Chambers
Royal Society Open Science
openscience@royalsociety.org

on behalf of Professor Chris Chambers (Registered Reports Editor, Royal Society Open Science)
openscience@royalsociety.org

Author's Response to Decision Letter for (RSOS-210641.R0)

See Appendix C.

RSOS-210641.R1 (Revision)

Review form: Reviewer 2

Is the language acceptable?

Yes

Do you have any ethical concerns with this paper?

No

Have you any concerns about statistical analyses in this paper?

No

Recommendation?

Accept with minor revision

Comments to the Author(s)

The results are interesting and mostly replicate those of the previous study. As they are relevant in the field of ToM in aging I would suggest commenting and elaborating them a little more in the discussion section.

Review form: Reviewer 4

Is the language acceptable?

Yes

Do you have any ethical concerns with this paper?

No

Have you any concerns about statistical analyses in this paper?

No

Recommendation?

Accept with minor revision

Comments to the Author(s)

There are just a handful of tyos:

On page 4, "believes" should be "beliefs"

On page 5, "That OA answered significant slower than YA..." should read "That OA answered significantly slower than YA..."

On page 9, "...an MRI-suitable keyboard that rested on there right thigh..." should read "...an MRI-suitable keyboard that rested on their right thigh..."

On page 9, "This speaks against the notion proposed by several authors (84,85) that age difference in perspective taking..." should read "This speaks against the notion proposed by several authors (84,85) that age differences in perspective taking..."

Decision letter (RSOS-210641.R1)

Dear Ms Stietz

On behalf of the Editor, I am pleased to inform you that your Stage 2 Replication submission RSOS-210641.R1 entitled "The Aging of the Social Mind: Replicating the preservation of socio-affective and the decline of socio-cognitive processes in old age" has been accepted for publication in Royal Society Open Science subject to minor revision in accordance with the referee suggestions. Please find the referees' comments at the end of this email.

The reviewers and Subject Editor have recommended publication, but also suggest some minor revisions to your manuscript. Therefore, I invite you to respond to the comments and revise your manuscript.

Please also ensure that all the below editorial sections are included where appropriate (a non-exhaustive example is included in an attachment):

- Ethics statement

- Data accessibility

If you wish to submit your supporting data or code to Dryad (<http://datadryad.org/>), or modify your current submission to dryad, please use the following link:
<http://datadryad.org/submit?journalID=RSOS&manu=RSOS-210641.R1>

- Competing interests

- Authors' contributions

- Acknowledgements

- Funding statement

Because the schedule for publication is very tight, it is a condition of publication that you submit the revised version of your manuscript within 7 days (i.e. by the 29-Jul-2021). If you do not think you will be able to meet this date please let me know immediately.

- 1) A text file of the manuscript (tex, txt, rtf, docx or doc), references, tables (including captions) and figure captions. Do not upload a PDF as your "Main Document".
- 2) A separate electronic file of each figure (EPS or print-quality PDF preferred (either format should be produced directly from original creation package), or original software format)
- 3) Included a 100 word media summary of your paper when requested at submission. Please ensure you have entered correct contact details (email, institution and telephone) in your user account
- 4) Included the raw data to support the claims made in your paper. You can either include your data as electronic supplementary material or upload to a repository and include the relevant DOI within your manuscript
- 5) Included your supplementary files in a format you are happy with (no line numbers, Vancouver referencing, track changes removed etc) as these files will NOT be edited in production

Kind regards,
Professor Chris Chambers
Royal Society Open Science
openscience@royalsociety.org

on behalf of Chris Chambers (Registered Reports Editor, Royal Society Open Science)
openscience@royalsociety.org

Associate Editor Comments to Author (Professor Chris Chambers):

Comments to the Author:

Two of the original Stage 1 reviewers kindly returned to evaluate the Stage 2 manuscript. As you will see, both are positive, finding that the primary Stage 2 criteria are met, and recommend acceptance following minor revision to correct typographic errors (Reviewer 4), and consider some minor addition to the Discussion (Reviewer 2). Based on my own reading, I feel the Discussion is sufficient as it stands for this article type, although I am happy for the authors to amend, if they wish, in accordance with the suggestion of Reviewer 2. Please respond to these points in a revision and full acceptance should be forthcoming without requiring further in-depth review.

Reviewers' comments to Author:

Reviewer: 2

Comments to the Author(s)

The results are interesting and mostly replicate those of the previous study. As they are relevant in the field of ToM in aging I would suggest commenting and elaborating them a little more in the discussion section.

Reviewer: 4

Comments to the Author(s)

There are just a handful of typos:

On page 4, "believes" should be "beliefs"

On page 5, "That OA answered significant slower than YA..." should read "That OA answered significantly slower than YA..."

On page 9, "...an MRI-suitable keyboard that rested on there right thigh..." should read "...an MRI-suitable keyboard that rested on their right thigh..."

On page 9, "This speaks against the notion proposed by several authors (84,85) that age difference in perspective taking..." should read "This speaks against the notion proposed by several authors (84,85) that age differences in perspective taking..."

Author's Response to Decision Letter for (RSOS-210641.R1)

See Appendix D.

Decision letter (RSOS-210641.R2)

Dear Julia,

It is a pleasure to accept your Stage 2 Replication entitled "The Aging of the Social Mind: Replicating the preservation of socio-affective and the decline of socio-cognitive processes in old age" in its current form for publication in Royal Society Open Science.

You can expect to receive a proof of your article in the near future. Please contact the editorial office (openscience@royalsociety.org) and the production office (openscience_proofs@royalsociety.org) to let us know if you are likely to be away from e-mail contact – if you are going to be away, please nominate a co-author (if available) to manage the proofing process, and ensure they are copied into your email to the journal.

on behalf of Professor Chris Chambers (Subject Editor)
openscience@royalsociety.org

Appendix A

Dear Professor Chris Chambers,

Thank you very much for giving us the chance to revise our manuscript. In revising we have carefully addressed each of the reviewers' comments and believe that our revision greatly benefitted from their suggestions.

In the following, the reviewers' comments are in italics, whereas our responses are in standard font. In the revised manuscript, we marked all changes in blue font colour for easier identification.

Please let us know if you have any further questions about our revision.

Kind regards,

Julia Stietz and Philipp Kanske
On behalf of all authors

Associate Editor Comments to Author (Professor Chris Chambers):

Associate Editor 1

Comments to the Authors:

*Four expert reviewers have now assessed the Stage 1 manuscript. All reviewers broadly judge **Stage 1 primary criterion #1** ("Whether the authors provide a sufficiently clear and detailed description of the methods") to met met, although there are still areas where adherence to this criterion can be improved (see comments of Reviewer 4, especially). As you will see, the reviews are more critical of **Stage 1 primary criterion #2** ("Whether the manuscript describes a sufficiently valid (i.e. close) and robust (e.g. statistically powerful) replication of the original study methods and rationale to provide an indication of replicability").*

The reviewers also request clarifications to the theoretical framing (Reviewer 3), terminology (Reviewer 2) and consideration of the validity of the design without additional measurements (Reviewer 1). In revising please note that unlike regular articles (or even Registered Reports), for the Replications article type at Royal Society Open Science, the key focus is on ensuring as close as possible a match between the methods of the original study and the replication, even when the methods of the original study may be suboptimal. Limitations in the design can then be addressed at Stage 2 in various ways (see <https://royalsocietypublishing.org/rsos/replication-studies> for details).

Response

We agree these are essential points. In revising we addressed the requests of the reviewers on the theoretical framing and terminology. We also took particular care in elaborating on the validity and robustness of our study (see below). We think the present study is indeed very close to the original experiment and statistically powerful enough to allow for testing the replicability of the previously observed effects. Our point-by-point responses to the reviewers' comments are detailed below.

Reviewer 1

We thank you very much for taking the time to read and comment on our manuscript. We address each of your concerns below. All changes made to the manuscript are marked in blue font colour in the revised manuscript.

Comment to the Authors:

I was surprised to see that the authors are not including confidence ratings (i.e., to index social metacognition) in this replication study. This is a limitation, given that, as Reiter et al. explain "social metacognition as an important facet of social information processing has yet to be studied from an adult developmental perspective.". To my knowledge, Reiter et al. (2017) is still the only study to have looked at social metacognition in aging to date, and they found that metacognition age deficits were evident in factual but not social reasoning – this is an important and interesting finding that needs to be replicated. The inclusion of the confidence/metacognition ratings in the current study would provide a much stronger contribution to the literature.

Response

Thanks a lot for pointing this out. We agree that an age deficit in metacognition on factual but not social reasoning is an important and interesting finding that needs to be replicated as well. In the present study, however, our goal was to replicate age differences in social affect (empathy and compassion) and social cognition (ToM). We also agree that not including the confidence ratings in the current study is a slight deviation from the original task. However, based on a multitude of previous evidence, we are confident that this does not affect the results for empathy, compassion, or ToM. We base our confidence on previous data from our lab showing that including the confidence rating at the end of the trial or leaving it out does not change the results of the previous measures within each trial (see e.g. the control group in the study by Winter et al. (2017; <https://doi.org/10.1038/s41598-017-00745-0>) that did not include a confidence rating, but found the same pattern of effects as other studies with that rating for empathy, compassion and ToM). Even if an entirely different rating (a prosociality rating on the "willingness to offer help") is included instead of the confidence rating, this does not change the results pattern for the other EmpaToM measures (see e.g. Lehman et al., in prep.; the results of the valence-ratings, compassion-ratings and the multiple-choice questions are attached at the end of this letter). For both studies (Lehmann et al. (in prep.) - replacing the confidence rating by a prosociality rating and Winter et al. (2017) - not including the confidence rating at all) the results of empathy, compassion and ToM are comparable to the results of Kanske et al. (2015; <https://doi.org/10.1016/j.neuroimage.2015.07.082>) and Reiter et al. (2017; <https://doi.org/10.1038/s41598-017-10669-4>). In sum, we think the evidence is very clear that changing or dropping the final (confidence) rating in the EmpaToM trials does not affect the previous measures that we aimed to replicate in the present study. To also clarify that to future readers of the manuscript, we added the following sentence in line 165 to 169:

"Other than Reiter and colleagues (59) we did not include a confidence rating after the multiple-choice questions, since we were especially interested in the effect of age on empathy, compassion, and ToM, rather than metacognition. Prior studies have shown that neither excluding the confidence rating at the end of the trial (37) nor replacing it by another rating (67) does affect the previous measures within each trial."

Comment to the Authors:

The authors define 'empathy' as 'feeling with somebody'. This is correct for affective empathy but not cognitive empathy. I think it would be helpful to explain that it is generally agreed that empathy is a multifaceted construct that has cognitive and affective components, and that in this study empathy is referring exclusively to the affective component of empathy. It might also be useful to mention that cognitive empathy overlaps with theory of mind.

Response

We agree with the reviewer that the term empathy is used quite differently in the field and that it is important to clearly specify the definition we apply here. We therefore added an elaborate explanation on this topic in line 33 to 45:

"Socio-affective processes include empathy and compassion. The definition of empathy ranges from a multifaceted construct - comprising affect sharing and empathic concern as an emotional and mentalizing or perspective-taking as a cognitive component of empathy (14-17) - to a very narrowly circumscribed construct -confining it to pure affect sharing (12,18). Here we apply the later definition that specifies empathy as feeling *with* somebody (18). More precisely empathy describes the sharing of another person's feeling while knowing that the other person's emotion is the cause of one's own feeling (18,19). Compassion, on the other hand, is defined as a positive, caring feeling *for* somebody that is strongly linked to the motivation to help the other person (19). Socio-cognitive processes include perspective-taking also known as mentalizing or Theory of Mind (ToM), which is defined as reasoning about the mental states (e.g., thoughts, desires, feelings) of another person (20,21). Some researchers also distinguish between an affective and cognitive component of ToM, with the former being the reasoning about emotions and the later one the reasoning about thoughts and believes (22). In the current study we mainly focus on cognitive ToM (in the following only referred to as ToM)."

Reviewer 2

We are grateful for your time and effort in evaluation our manuscript. We address your concern below. All changes are marked in blue font colour for easier identification in the revised manuscript.

Comment to the Authors:

My only concern regards the use of the word "naturalistic" as definition of the EmpaToM task. I'd suggest to use ecological task instead of naturalistic task.

Response

We changed "a naturalistic paradigm" to "**an ecologically valid paradigm**" in line 81.

Reviewer 3

We thank you very much for taking the time to read and comment on our manuscript. We took great care to address your concerns. All changes are marked in blue font colour for easier identification in the revised manuscript.

Comment to the Authors:

Introduction: For a study examining social cognition and ageing, the authors severely underrepresent the vast and interesting literature on this topic. A reader might think this is an under researched area, but it is not, and this needs to be addressed.

Response

Thank you for pointing this out. We agree that the literature on social cognition and ageing is very interesting and now elaborate more on previous work in the field. Specifically, we included an additional paragraph dedicated to a more comprehensive literature review in line 60 to 67:

“Over the last 30 years a vast body of research emerged on this topic (38). One of the first studies examining the effect of age on social cognition (39) revealed a preservation or even enhancement of ToM in older compared to younger adults. This finding was followed by an increasing number of studies showing the opposite effect, an age-related decline in taking the perspective of others (40). However, there are still studies showing a preservation of socio-cognitive processes with age (41-47). Research on adult age differences in social affect is a little more consistent with most studies showing a preservation or enhancement in empathy and compassion (48-51) and only a few a decline (52-54).”

Comment to the Authors:

The socio-emotional selective theory by Carstensen et al. is considered a theory of motivation and attention in ageing, to my knowledge it does not state 'that cognitive components decline with age but emotional components are preserved or even enhance with age' as the authors argue in their introduction. I would recommend some clarification here and carefully consider the relationship between social cognition and attentional/motivation theories of ageing as they are quite different.

Response

Thanks a lot for your suggestion. We agree that our statement about the socioemotional selectivity theory by Carstensen et al. (1999) was not as clearly described as it should have been. To better explain and clarify our thoughts on the relation between the shift of motivation proposed by Carstensen et al. (1999) and the development of socio-affective and -cognitive processes with age, we added an additional paragraph on this in line 67 to 77:

“Intact or enhanced social affect in older relative to younger adults might be explained by the assumption that emotional goals (e.g., paying attention to emotions) are considered as more relevant and important, and thus are increasingly more selected and pursued as people age (55). Whereas a decreased social cognition might be explained by the assumption that knowledge-related goals (e.g., pursuit information about the social world) are considered as less valuable and are selected and pursued less likely which is proposed by the socioemotional selectivity theory of Carstensen and colleagues (55). The authors claim that age-related differences in selectively pursuing social goals may arise from altered perception of the remaining available lifetime. Thus, the socioemotional selectivity theory suggests a shift of motivation from knowledge-related goals to emotional goals from early adulthood to old age in the consequence of a perception of ascendingly constraint time with increasing chronological age (55).”

We further amended the following sentences in line 77 to 86:

“However, most adult development studies until recently focused on the effect of age on *either* the socio-emotional *or* the socio-cognitive route (51,56–58), which precluded the direct comparison of the lifespan development of social affect and cognition. One exception is the study by Reiter and colleagues (59). The authors investigated age-related differences in empathy, compassion, and ToM with an ecological valid paradigm (EmpaToM;25) that enables measuring all three abilities within the same person based on the same stimuli. They found that older adults (OA) showed the same level of empathy and increased compassion compared to younger adults (YA). In contrast to that, YA outperformed OA in ToM. Thus, Reiter and colleagues (59) observed a preservation or even enhancement for socio-affective processes, but a decline in socio-cognitive processes in old age using the EmpaToM, which is in line with the suggestion of the socio-emotional selectivity theory.”

Reviewer 4

Thank you so much for your thoughtful comments on our manuscript. We carefully address each of them below. In the revised manuscript, all changes are marked in blue font colour for easier identification.

Comment to the Authors:

The authors discuss theory of mind (ToM) as one component of social cognition. However, as we know, ToM can be subdivided into affective and cognitive ToM. It is a little unclear how this relates to the socio-affective and socio-cognitive processes they discuss in their paper. I think it's important to explain this within the study and be clear how it relates to the processes they are investigating, especially as age effects are consistently found in the aging literature on cognitive ToM but less consistently so for affective ToM. I realize Reiter et al. did not do this but I think it would improve how these findings relate to the social cognition literature.

Response

Thank you for the suggestion. We agree that it is important to explain clearly which component of ToM is investigated in our study. We added explanatory remarks on the differentiation of ToM into its affective and cognitive components and how this relates to the socio-cognitive process we examine in our study (line 42 to 45):

“Some researchers also distinguish between an affective and cognitive component of ToM, with the former being the reasoning about emotions and the later one the reasoning about thoughts and believes (22). In the current study we mainly focus on cognitive ToM (in the following only referred to as ToM).”

To clarify the methods section of the manuscript, we further added “cognitive” and replaced “mental state” by “thoughts” in the following sentence (line 152-155):

“The EmpaToM (25) is a video-based social interaction task that enables measuring empathy (as the sharing of a narrator’s emotion), compassion (as concern for the narrator) and **cognitive** ToM (via the response to a multiple-choice question about the **thoughts** of the narrator) within the same person in a single task (Fig. 1).”

Comment to the Authors:

It is also not clear from the EmpaToM Task which aspect of ToM it is assessing. From the methods, it appears to be examining cognitive rather than affective ToM but this needs some clarity.

Response

Thank you for pointing this out. You are completely right that it is "cognitive ToM" that the task assess. We now clarified this in the methods section of the manuscript by adding "cognitive" and replacing "mental state" by "thoughts" in the following sentence (line 152-155):

"The EmpaToM (25) is a video-based social interaction task that enables measuring empathy (as the sharing of a narrator's emotion), compassion (as concern for the narrator) and **cognitive** ToM (via the response to a multiple-choice question about the **thoughts** of the narrator) within the same person in a single task (Fig. 1)."

Comment to the Authors:

I did wonder whether the empathy measure was assessing affective ToM but the questions suggest that participants indicated how they felt rather than the speaker in the video.

Response

Thank you for asking. You are right that the empathy measure in our task assess "emotional empathy" or "affect sharing" rather than "affective ToM". We describe that in the methods section: the EmpaToM measures "empathy (as the sharing of a narrator's emotion)" -line 152 to 153- with the participants reporting "how they feel themselves (empathy measure, 7 sec.) on a visual analogue scale ranging from positive to negative" -line 157 to 158-. The logic is that negative emotions after an emotionally negative video indicate that the narrator's emotion was empathically shared.

Comment to the Authors:

The participants in the current replication were studied inside the scanner. It is not clear to me why this was necessary and I do have concerns about the change in the testing environment compared to Reiter et al. This change in the environment might have an influence on older adults' performance as the scanner could be conceived as a stressful, unusual environment. Were these participants used to being inside a scanner? If you fail to replicate, might this be why?

Response

Thanks a lot for pointing this out. The practical reason is that this replication study was embedded in a larger project that caused the change in the testing environment. We agree that changing the testing environment could in principle influence the performance on a task one wants to replicate. However, we consider the risk in our particular case, that is testing the EmpaToM inside a scanner, as extremely low. First, the EmpaToM has control conditions. The accuracy and reaction time on multiple-choice questions that require ToM can be compared to the accuracy and reaction time on multiple-choice questions that require factual reasoning (control condition). The same applies for the valence ratings (empathy measure) for which emotionally negative videos and emotionally neutral videos (control condition) can be compared. Any general effects of the test environment should show in the experimental and the control conditions alike and, thus, render the experimental effects interpretable. Second, the EmpaToM is a well validated task that has already been used in- and outside the MRI scanner in several studies which yielded comparable patterns of behavioural results (see e.g. outside a scanner - control group of Winter et al., 2017; <https://doi.org/10.1038/s41598-017->

[00745-0](https://doi.org/10.1016/j.neuroimage.2015.07.082) compared to inside a scanner - the sample of Kanske et al., 2015; <https://doi.org/10.1016/j.neuroimage.2015.07.082>). Also the group of younger participants in the study by Reiter et al. (2017; <https://doi.org/10.1038/s41598-017-10669-4>) done outside the scanner showed a comparable pattern of behaviour as the sample of Kanske et al. (2015) that performed the EmpaToM in an MRI scanner. Third, the familiarity with the testing environment should not only affect the older adult's but also the younger adult's performance, since there is no reason why our younger participants should have been more familiar with an MRI scanner in our study than the older participants. [In fact, one could speculate whether older participants might not have had a greater chance of getting familiar with being in an MRI scanner throughout their lifespan, especially due to medical diagnostic uses of MRI scanners.] Furthermore, a study by Förster et al. (in prep.) indicates that the familiarity with the scanner environment does not change the results of the EmpaToM. The authors found good test-retest-reliability of the EmpaToM even when the participants returned to the scanner four times to perform the task (intraclass correlations range from .65 to .78 across the four measurements). Fourth, the EmpaToM provides very robust findings in general. Tholen et al. (2020) just recently showed with a behavioural and fMRI item analysis in two independent samples of participants that the EmpaToM yields stable and reproducible results. We are therefore confident that the change in the testing environment should not affect our replication endeavour but will add a section on this issue in the discussion.

Comment to the Authors:

Having an equivalent level of education in the younger and older groups is certainly unusual. Why might this have occurred? How did the education levels compare with Reiter et al.? This is important because studies have shown that higher education protects against the impact of healthy and pathological aging. Similarly, it's a little unusual for older adults not to be impaired in terms of speed. If the authors fail to replicate, might it be something related to their slight positive selection bias?

Response

Thanks a lot for your question and suggestion. Our sample and Reiter et al. (2017)'s sample are in fact comparable with regard to the educational level. The younger and older adults of Reiter et al. (2017) did not differ in their years of education as is the case for the present study. Nevertheless, Reiter and colleagues found an age-related decline in social cognition, but not social affect which makes it even more interesting to replicate their findings.

To avoid confusion on the comparability of our and Reiter et al. (2017), we changed the word "similar" to "comparable" in the following sentence (line 136-138): "YA and OA did also not differ significantly in their years of education, relationship status or residence (see table 1 for descriptive and inferential statistics) comparable to Reiter and colleagues' sample (59)."

We further added "and Reiter and colleagues' sample (59)" to the following sentence (line 138-140): "However, this might hint to a slight positive selection bias for the OA in our sample and Reiter and colleagues' sample (59), since it is not common for OA to have an equivalent level of education as YA in aging studies."

Regarding the impairment in cognitive speed, the older adults of our sample were slower in answering the IDP, but not less accurate. Thus, they were impaired in one aspect of the cognitive speed measure (reaction time) like Reiter et al. (2017)'s sample or other population-based lifespan samples (Li et al., 2004) but not on accuracy as assessed by the IDP task.

To clarify this finding and how it relates to Reiter et al. (2017)'s sample as well as the literature we revised two sentences (line 177-181):

"That OA answered significant slower than YA on the IDP is in line with Reiter and colleagues' sample (59) as well as findings of a population-based lifespan sample (72). However, that OA answered as accurately as YA on the IDP is not in line with Reiter and colleagues' sample (59) and findings of a population-based lifespan sample (72) which again hints to a slight positive selection bias for the OA in our sample."

We further added "accuracy" in the following sentence (line 182-184): "Thus, regarding cognitive abilities the current sample matched the sample of Reiter and colleagues (59) with comparable means and test statistics on the cognitive measures, except for cognitive speed (IDP accuracy)."

Additional to that, we will address the characteristics of our sample and how they relate to our findings again in the discussion section of the manuscript.

Comment to the Authors:

Are there issues with the EmpaToM Task having self-report measures for the empathy and compassion questions but an objective measure of ToM?

Response

The reviewer is correct that the EmpaToM, as other tasks assessing social affect (Dziobek et al., 2008; Klimecki et al., 2013), measures empathy and compassion as self-reports on visual analogue scales, while it measures social cognition (ToM) more objectively as a performance measure in multiple-choice questions. This is based on the idea, that empathy and compassion are socio-emotional processes that by their very nature are subjective contents of experience. Thus, the question how it feels to experience empathy and compassion can only be answered through self-reports on the behavioural level. That is the nature of the beast. The strength of the EmpaToM is that it measures empathy and compassion directly after an emotional experience. This enhances the chance to measure the actual feeling more accurately than by self-report questionnaires in which the participants must reconstruct and synthesize retrospectively. Furthermore, the measures of social affect that are assessed by the EmpaToM seem to be not influenced by social desirability, since Winter et al. (2017) did find a preserved ToM but decreased empathy in their sample.

Comment to the Authors:

What happened if participants didn't respond within the time limits? Was that item excluded and the score proportionally adjusted or did participants receive a score of zero?

Response

Thanks a lot for asking and making us aware that we missed to describe that in detail in the manuscript. Trials in which participants did not respond on all three measures (valence rating, concern rating and multiple-choice question), so called sleep trials, were excluded from the whole analysis. For the calculation of the reaction time on ToM and factual reasoning questions we only used correct items, excluding false response and misses and adjusting the score proportionally. For the calculation of the accuracy on ToM and factual reasoning questions we set misses to zero.

To address your comment, we clarified our methods section by adding the following sentences in line 195 to 201:

“For the calculation of the mean RT of ToM/nonToM questions only correct items (excluding false response and no response items) were used. For the calculation of the accuracy of ToM/nonToM questions no response items were set to zero. Due to a few technical problems with the button box that had been stuck for a few trials while the EmpaToM was performed, these trials were excluded from all measures, as were sleep trials in which participants did not respond in time on all three measures (valence rating, concern rating and multiple-choice question). The empathy, compassion and ToM scores were proportionally adjusted for these trials.”

We additionally removed “of all correct trials” and changed “ACC” to “accuracy” in the following sentence (line 192 to 195):

“Analogous to Reiter and colleagues (59) measures of empathy (mean valence rating for the emotional and neutral condition separately), compassion (mean concern rating across all conditions) and ToM/nonToM (mean reaction time (RT) ~~of all correct trials~~ and error rates (**accuracy**) for the ToM and nonToM condition separately) were derived from the EmpaToM.”

Comment

It is unclear why fluid and verbal measures were considered in separate ANCOVAs. I may have misunderstood but I think Reiter et al considered them in the same analyses?

Response

Thank you for asking. This was actually double-checked with the authors of Reiter et al. (2017), before specifying our analysis plan. They indeed performed two independent ANCOVAs, one with the fluid proxy as covariate and the other with the verbal proxy as covariate.

Comment

Were any mood measures included as this might influence performance? I know that Reiter administered the PANAS.

Response

Thank you for asking. In the Reiter et al 2017 study, the PANAS was administered as part of a general battery for the purpose of sample characterization without any direct hypothesis relates to EmpaToM measures. Here, due to this lack of hypotheses with respect to the PANAS, we did not include the PANAS or any other mood measures, in order to minimize the burden for our participants as far as possible, as this is generally higher in (f)MRI studies including preparation for the scanner etc.

The reviewer is absolutely right that general mood/affect might have an impact on our EmpaToM measures, particularly on empathy as this is indeed operationalized by administering valence ratings. Note that this is why control for the effect of mood on empathy directly within the task: The EmpaToM includes valence ratings of emotionally negative and neutral videos where the later ones actually serve as a control condition reflecting baseline affect. If participants were already in a negative or positive mood prior to performing the task this would affect and be depicted in the valence ratings of the neutral videos (as a deviation from neutral). By analysing an interaction effect of video type (emotional vs. neutral) by age group as reflecting age differences with respect to empathy (compare Reiter et al., 2017) the effect of

general mood/affect is controlled for. We amended the methods section to make this clearer in line 208 to 210:

“By analysing the interaction of video type (emotional vs. neutral) and age group (YA vs. OA) to assess age differences in empathy the effect of general mood is controlled for, since the valence ratings of the neutral videos serve as control conditions that reflect baseline affect.”

References used in this letter of response

- Carstensen, L. L., Isaacowitz, D. M., & Charles, S. T. (1999). Taking time seriously. A theory of socioemotional selectivity. *The American Psychologist*, *54*(3), 165–181. <https://doi.org/10.1037//0003-066x.54.3.165>
- Dziobek, I., Rogers, K., Fleck, S., Bahnemann, M., Heekeren, H. R., Wolf, O. T., & Convit, A. (2008). Dissociation of Cognitive and Emotional Empathy in Adults with Asperger Syndrome Using the Multifaceted Empathy Test (MET). *Journal of Autism and Developmental Disorders*, *38*(3), 464–473. <https://doi.org/10.1007/s10803-007-0486-x>
- Förster, K., Lehmann, K., Tholen, M., Trautwein, F.-M., Böckler, A., Singer, A., & Kanske, P. Test-retest reliability of empathy and theory of mind [in preparation]. Department of Psychology, Technische Universität Dresden
- Kanske, P., Böckler, A., Trautwein, F.-M., & Singer, T. (2015). Dissecting the social brain: Introducing the EmpaToM to reveal distinct neural networks and brain–behavior relations for empathy and Theory of Mind. *NeuroImage*, *122*, 6–19. <https://doi.org/10.1016/j.neuroimage.2015.07.082>
- Klimecki, O. M., Leiberg, S., Lamm, C., & Singer, T. (2013). Functional Neural Plasticity and Associated Changes in Positive Affect After Compassion Training. *Cerebral Cortex*, *23*(7), 1552–1561. <https://doi.org/10.1093/cercor/bhs142>
- Lehmann, K., Böckler, A., Klimecki, O., Müller-Liebmann, C., & Kanske, P. Feeling with or thinking as: How empathy and mentalizing shape prosociality [in preparation]. Department of Psychology, Technische Universität Dresden
- Li, S.-C., Lindenberger, U., Hommel, B., Aschersleben, G., Prinz, W., & Baltes, P. B. (2004). Transformations in the couplings among intellectual abilities and constituent cognitive processes across the life span. *Psychological Science*, *15*(3), 155–163. <https://doi.org/10.1111/j.0956-7976.2004.01503003.x>
- Reiter, A. M. F., Kanske, P., Eppinger, B., & Li, S.-C. (2017). The Aging of the Social Mind—Differential Effects on Components of Social Understanding. *Scientific Reports*, *7*(1), 11046. <https://doi.org/10.1038/s41598-017-10669-4>
- Tholen, M. G., Trautwein, F.-M., Böckler, A., Singer, T., & Kanske, P. (2020). Functional magnetic resonance imaging (fMRI) item analysis of empathy and theory of mind. *Human Brain Mapping*, *41*(10), 2611–2628. <https://doi.org/10.1002/hbm.24966>
- Winter, K., Spengler, S., Bermpohl, F., Singer, T., & Kanske, P. (2017). Social cognition in aggressive offenders: Impaired empathy, but intact theory of mind. *Scientific Reports*, *7*(1), 670. <https://doi.org/10.1038/s41598-017-00745-0>

References of the revised manuscript

1. Harper S. Economic and social implications of aging societies. *Science*. 2014 Oct 31;346(6209):587–91.
2. United Nations. World Population Prospects - Population Division [Internet]. 2017 [cited 2020 Dec 7]. Available from: <https://population.un.org/wpp/>
3. World Health Organization. Health and Aging. [Internet]. 2011 [cited 2020 Dec 7]. Available from: https://www.who.int/ageing/publications/global_health.pdf
4. Fotenos. Brain Volume Decline in Aging: Evidence for a Relation Between Socioeconomic Status, Preclinical Alzheimer Disease, and Reserve. *ARCH NEUROL*. 2008;65(1):8.
5. Jenkin CR, Eime RM, Westerbeek H, O'Sullivan G, van Uffelen JGZ. Sport and ageing: a systematic review of the determinants and trends of participation in sport for older adults. *BMC Public Health* [Internet]. 2017 Dec 22 [cited 2020 Nov 17];17. Available from: <https://www.ncbi.nlm.nih.gov/pmc/articles/PMC5741887/>
6. Lee C, Longo V. Dietary restriction with and without caloric restriction for healthy aging. *F1000Res* [Internet]. 2016 Jan 29 [cited 2020 Nov 17];5. Available from: <https://www.ncbi.nlm.nih.gov/pmc/articles/PMC4755412/>
7. McIntyre RL, Daniels EG, Molenaars M, Houtkooper RH, Janssens GE. From molecular promise to preclinical results: HDAC inhibitors in the race for healthy aging drugs. *EMBO Molecular Medicine*. 2019 Sep 1;11(9):e9854.
8. Passow S, Thurm F, Li S-C. Activating Developmental Reserve Capacity Via Cognitive Training or Non-invasive Brain Stimulation: Potentials for Promoting Fronto-Parietal and Hippocampal-Striatal Network Functions in Old Age. *Front Aging Neurosci* [Internet]. 2017 [cited 2020 Dec 7];9. Available from: <https://www.frontiersin.org/articles/10.3389/fnagi.2017.00033/full>
9. Willcox BJ, Willcox DC. Caloric Restriction, CR Mimetics, and Healthy Aging in Okinawa: Controversies and Clinical Implications. *Curr Opin Clin Nutr Metab Care*. 2014 Jan;17(1):51–8.
10. Holt-Lunstad, Smith, Layton. Social Relationships and Mortality Risk: A Meta-analytic Review. 2010.
11. Lövdén M, Ghisletta P, Lindenberger U. Social participation attenuates decline in perceptual speed in old and very old age. *Psychology and Aging*. 2005;20(3):423–34.
12. Kanske P. The social mind: disentangling affective and cognitive routes to understanding others. *Interdisciplinary Science Reviews*. 2018 Jun;43(2):115–24.
13. Kanske P, Böckler A, Singer T. Models, Mechanisms and Moderators Dissociating Empathy and Theory of Mind. In: Wöhr M, Krach S, editors. *Social Behavior from Rodents to Humans: Neural Foundations and Clinical Implications* [Internet]. Cham: Springer International Publishing; 2017 [cited 2019 Jan 18]. p. 193–206. (Current Topics in Behavioral Neurosciences). Available from: https://doi.org/10.1007/7854_2015_412
14. Dziobek I, Rogers K, Fleck S, Bahnemann M, Heekeren HR, Wolf OT, et al. Dissociation of Cognitive and Emotional Empathy in Adults with Asperger Syndrome Using the Multifaceted Empathy Test (MET). *J Autism Dev Disord*. 2008 Mar 1;38(3):464–73.
15. Blanke ES, Riediger M. Reading thoughts and feelings in other people: Empathic accuracy across adulthood. *Prog Brain Res*. 2019;247:305–27.
16. Zaki J. Moving beyond Stereotypes of Empathy. *Trends in Cognitive Sciences*. 2017 Feb;21(2):59–60.
17. Wieck C, Kunzmann U. Age differences in empathy: Multidirectional and context-dependent. *Psychology and Aging*. 2015 Jun;30(2):407–19.

18. Singer T, Klimecki OM. Empathy and compassion. *Current Biology*. 2014 Sep;24(18):R875–8.
19. de Vignemont F, Singer T. The empathic brain: how, when and why? *Trends in Cognitive Sciences*. 2006 Oct;10(10):435–41.
20. Frith C, Frith U. Theory of mind. *Curr Biol*. 2005 Sep 6;15(17):R644–646.
21. Wimmer H, Perner J. Beliefs about beliefs: Representation and constraining function of wrong beliefs in young children’s understanding of deception. *Cognition*. 1983;13(1):103–28.
22. Healey ML, Grossman M. Cognitive and Affective Perspective-Taking: Evidence for Shared and Dissociable Anatomical Substrates. *Front Neurol* [Internet]. 2018 Jun 25 [cited 2019 Jan 18];9. Available from: <https://www.ncbi.nlm.nih.gov/pmc/articles/PMC6026651/>
23. Stietz J, Jauk E, Krach S, Kanske P. Dissociating Empathy From Perspective-Taking: Evidence From Intra- and Inter-Individual Differences Research. *Front Psychiatry*. 2019 Mar 14;10:126.
24. Kanske P, Böckler A, Trautwein F-M, Parianen Lesemann FH, Singer T. Are strong empathizers better mentalizers? Evidence for independence and interaction between the routes of social cognition. *Soc Cogn Affect Neurosci*. 2016 Sep;11(9):1383–92.
25. Kanske P, Böckler A, Trautwein F-M, Singer T. Dissecting the social brain: Introducing the EmpaToM to reveal distinct neural networks and brain–behavior relations for empathy and Theory of Mind. *NeuroImage*. 2015 Nov;122:6–19.
26. Lamm C, Decety J, Singer T. Meta-analytic evidence for common and distinct neural networks associated with directly experienced pain and empathy for pain. *NeuroImage*. 2011 Feb 1;54(3):2492–502.
27. Lamm C, Silani G, Singer T. Distinct neural networks underlying empathy for pleasant and unpleasant touch. *Cortex*. 2015 Sep;70:79–89.
28. Schurz M, Radua J, Tholen MG, Maliske L, Margulies DS, Mars RB, et al. Toward a hierarchical model of social cognition: A neuroimaging meta-analysis and integrative review of empathy and theory of mind. *Psychological Bulletin* [Internet]. 2020;1105 [cited 2020 Dec 4]; Available from: <https://psycnet.apa.org/fulltext/2020-82377-001.pdf>
29. Bzdok D, Schilbach L, Vogeley K, Schneider K, Laird AR, Langner R, et al. Parsing the neural correlates of moral cognition: ALE meta-analysis on morality, theory of mind, and empathy. *Brain Structure and Function*. 2012 Oct;217(4):783–96.
30. Schurz M, Radua J, Aichhorn M, Richlan F, Perner J. Fractionating theory of mind: A meta-analysis of functional brain imaging studies. *Neuroscience & Biobehavioral Reviews*. 2014 May;42:9–34.
31. Baron-Cohen S. Theory of mind and autism: A review. In: *International Review of Research in Mental Retardation* [Internet]. Academic Press; 2000 [cited 2020 Dec 4]. p. 169–84. (Autism; vol. 23). Available from: <http://www.sciencedirect.com/science/article/pii/S0074775000800105>
32. Fletcher-Watson S, Bird G. Autism and empathy: What are the real links? *Autism*. 2020 Jan;24(1):3–6.
33. Hadjikhani N, Zürcher NR, Rogier O, Hippolyte L, Lemonnier E, Ruest T, et al. Emotional contagion for pain is intact in autism spectrum disorders. *Translational Psychiatry*. 2014 Jan;4(1):e343–e343.
34. Smith A. The Empathy Imbalance Hypothesis of Autism: A Theoretical Approach to Cognitive and Emotional Empathy in Autistic Development. *Psychol Rec*. 2009 Jul 1;59(3):489–510.
35. Trimmer E, McDonald S, Rushby JA. Not knowing what I feel: Emotional empathy in autism spectrum disorders. *Autism*. 2017 May 1;21(4):450–7.
36. Preckel K. On the interaction of social affect and cognition: empathy, compassion and theory of mind. *Current Opinion in Behavioral Sciences*. 2018;6.

37. Winter K, Spengler S, Bermpohl F, Singer T, Kanske P. Social cognition in aggressive offenders: Impaired empathy, but intact theory of mind. *Sci Rep*. 2017 06;7(1):670.
38. Stietz J, Kanske P. The Aging of the social mind: A Systematic Review. in prep.;
39. Happe F, Winner E, Brownell H. The getting of wisdom: Theory of mind in old age. *DEVELOPMENTAL PSYCHOLOGY*. 1998 Mar;34(2):358–62.
40. Henry JD, Phillips LH, Ruffman T, Bailey PE. A Meta-Analytic Review of Age Differences in Theory of Mind. *Psychol Aging*. 2013 Sep;28(3):826–39.
41. Girardi A, Sala SD, MacPherson SE. Theory of mind and the Ultimatum Game in healthy adult aging. *Experimental Aging Research*. 2018 May 27;44(3):246–57.
42. Grainger SA, Henry JD, Naughtin CK, Comino MS, Dux PE. Implicit false belief tracking is preserved in late adulthood. *Quarterly Journal of Experimental Psychology*. 2018 Sep;71(9):1980–7.
43. Hughes C, Cassidy BS, Faskowitz J, Avena-Koenigsberger A, Sporns O, Krendl AC. Age differences in specific neural connections within the Default Mode Network underlie theory of mind. *NeuroImage*. 2019 May 1;191:269–77.
44. Lecce S, Ceccato I, Cavallini E. Investigating ToM in aging with the MASC: from accuracy to error type. *Aging, Neuropsychology, and Cognition*. 2019 Jul 4;26(4):541–57.
45. Slessor G, Phillips LH, Bull R. Exploring the specificity of age-related differences in theory of mind tasks. *Psychology and Aging*. 2007 Sep;22(3):639–43.
46. Baksh RA, Abrahams S, Auyeung B, MacPherson SE. The Edinburgh Social Cognition Test (ESCoT): Examining the effects of age on a new measure of theory of mind and social norm understanding. van den Bos R, editor. *PLOS ONE*. 2018 Apr 17;13(4):e0195818.
47. Castelli I, Baglio F, Blasi V, Alberoni M, Falini A, Liverta-Sempio O, et al. Effects of aging on mindreading ability through the eyes: An fMRI study. *Neuropsychologia*. 2010 Jul 1;48(9):2586–94.
48. Bailey PE, Henry JD, Von Hippel W. Empathy and social functioning in late adulthood. *Aging & Mental Health*. 2008 Jul;12(4):499–503.
49. Hühnel I, Fölster M, Werheid K, Hess U. Empathic reactions of younger and older adults: No age related decline in affective responding. *Journal of Experimental Social Psychology*. 2014 Jan;50:136–43.
50. Richter D, Kunzmann U. Age differences in three facets of empathy: Performance-based evidence. *Psychology and Aging*. 2011 Mar;26(1):60–70.
51. Sze JA, Gyurak A, Goodkind MS, Levenson RW. Greater emotional empathy and prosocial behavior in late life. *Emotion*. 2012 Oct;12(5):1129–40.
52. Chen Y-C, Chen C-C, Decety J, Cheng Y. Aging is associated with changes in the neural circuits underlying empathy. *Neurobiology of Aging*. 2014 Apr 1;35(4):827–36.
53. Phillips LH, MacLean RDJ, Allen R. Age and the Understanding of Emotions: Neuropsychological and Sociocognitive Perspectives. *The Journals of Gerontology Series B: Psychological Sciences and Social Sciences*. 2002 Nov 1;57(6):P526–30.
54. Schieman S, Gundy KV. The Personal and Social Links between Age and Self-Reported Empathy. *Social Psychology Quarterly*. 2000 Jun;63(2):152.
55. Carstensen LL, Isaacowitz DM, Charles ST. Taking time seriously. A theory of socioemotional selectivity. *Am Psychol*. 1999 Mar;54(3):165–81.
56. Bailey PE, Henry JD. Growing Less Empathic With Age: Disinhibition of the Self-Perspective. *The Journals of Gerontology Series B: Psychological Sciences and Social Sciences*. 2008 Jul 1;63(4):P219–26.

57. Calso C, Besnard J, Allain P. Study of the theory of mind in normal aging: focus on the deception detection and its links with other cognitive functions. *Aging, Neuropsychology, and Cognition*. 2019 Jun 12;0(0):1–23.
58. Sze JA. Understanding the Emotions of Others: Loss or Gain in Aging? [Internet]. UC Berkeley; 2010 [cited 2019 Mar 20]. Available from: <https://escholarship.org/uc/item/1z245973>
59. Reiter AMF, Kanske P, Eppinger B, Li S-C. The Aging of the Social Mind - Differential Effects on Components of Social Understanding. *Scientific Reports*. 2017 Sep 8;7(1):11046.
60. Ioannidis JPA. Why Most Published Research Findings Are False. *PLoS Medicine*. 2005 Aug 30;2(8):e124.
61. Open Science Collaboration. Estimating the reproducibility of psychological science. *Science*. 2015 Aug 28;349(6251):aac4716–aac4716.
62. Camerer CF, Dreber A, Holzmeister F, Ho T-H, Huber J, Johannesson M, et al. Evaluating the replicability of social science experiments in *Nature* and *Science* between 2010 and 2015. *Nature Human Behaviour*. 2018 Sep;2(9):637–44.
63. Klein RA, Vianello M, Hasselman F, Adams BG, Adams RB, Alper S, et al. Many Labs 2: Investigating Variation in Replicability Across Samples and Settings. *Advances in Methods and Practices in Psychological Science*. 2018 Dec;1(4):443–90.
64. Tholen MG, Trautwein F-M, Böckler A, Singer T, Kanske P. Functional magnetic resonance imaging (fMRI) item analysis of empathy and theory of mind. *Human Brain Mapping*. 2020;41(10):2611–28.
65. Nasreddine ZS, Phillips NA, Bédirian V, Charbonneau S, Whitehead V, Collin I, et al. The Montreal Cognitive Assessment, MoCA: a brief screening tool for mild cognitive impairment. *J Am Geriatr Soc*. 2005 Apr;53(4):695–9.
66. Davis DH, Creavin ST, Yip JL, Noel-Storr AH, Brayne C, Cullum S. Montreal Cognitive Assessment for the diagnosis of Alzheimer’s disease and other dementias. Cochrane Dementia and Cognitive Improvement Group, editor. *Cochrane Database of Systematic Reviews* [Internet]. 2015 Oct 29 [cited 2020 Dec 7]; Available from: <http://doi.wiley.com/10.1002/14651858.CD010775.pub2>
67. Lehmann K, Böckler A, Klimecki O, Müller-Liebmann C, Kanske P. Feeling with or thinking as: How empathy and mentalizing shape prosociality. in prep.
68. Reitan RM. The relation of the trail making test to organic brain damage. *J Consult Psychol*. 1955 Oct;19(5):393–4.
69. Ekstrom, French, Harman, Dermen. *Manual for Kit of Factor- referenced Cognitive Tests*. 1976;
70. *Decision Making Individual Differences Inventory - Wechsler Adult Intelligence Scale* [Internet]. [cited 2020 Aug 24]. Available from: http://www.sjdm.org/dmidi/Wechsler_Adult_Intelligence_Scale.html
71. Lindenberger, Mayr, Reinhold. *Speed and Intelligence in Old Age*. 1993;
72. Li S-C, Lindenberger U, Hommel B, Aschersleben G, Prinz W, Baltes PB. Transformations in the couplings among intellectual abilities and constituent cognitive processes across the life span. *Psychol Sci*. 2004 Mar;15(3):155–63.
73. Thurm F, Zink N, Li S-C. Comparing Effects of Reward Anticipation on Working Memory in Younger and Older Adults. *Front Psychol* [Internet]. 2018 [cited 2020 Dec 8];9. Available from: <https://www.frontiersin.org/articles/10.3389/fpsyg.2018.02318/full#B51>
74. R Core Team. *R: A language and environment for statistical computing*. 2020;

75. Kassambara A. Pipe-Friendly Framework for Basic Statistical Tests [R package rstatix version 0.6.0] [Internet]. Comprehensive R Archive Network (CRAN); 2020 [cited 2020 Nov 17]. Available from: <https://CRAN.R-project.org/package=rstatix>
76. Mair P, Wilcox R. Robust statistical methods in R using the WRS2 package. Behav Res. 2020 Apr 1;52(2):464–88.
77. The Math Works, Inc. MATLAB (Version 2019a) [Internet]. 2019. Available from: <https://www.mathworks.com/>

Results of Lehmann et al. (in prep.)

Appendix B

Dear Professor Chris Chambers,

Thank you very much for giving us the chance to revise our manuscript. We carefully addressed the concerns of reviewer 1 and believe that the manuscript is now very specific and explicit about the partial nature of the replication.

In the following, the reviewers' comments are in italics, whereas our responses are in standard font. In the revised manuscript, we marked all changes in blue font colour for easier identification.

Please let us know if you have any further questions about our revision.

Kind regards,

Julia Stietz and Philipp Kanske
On behalf of all authors

Associate Editor (Professor Chris Chambers):

Comments to the Authors:

Two of the four reviewers who assessed the first submission kindly returned to assess the revised Stage 1 manuscript. As you will see, Reviewer 4 is now satisfied and recommends IPA, whereas Reviewer 1 remains concerned about the omission of confidence ratings, and on this basis judges that Stage 1 primary criterion #2 is unmet (including especially the methodological proximity of the replication to the original study).

Meeting the primary criteria is important for Stage 1 Replications, and so I have looked closely at the reviewer's concern, the revised manuscript, and the authors' previous response to this point. On balance, while I do share the reviewer's concern, I think that even in spite of this omission, the replication is sufficiently close to the original study for the research to be partially informative about replicability, and I have therefore decided not to reject the manuscript on this basis. However, as the reviewer suggests, I would like the authors to be more explicit about the partial nature of the replication.

In revising, please also address the other concern raised by Reviewer 1 concerning the inclusion criteria.

Provided the authors are able to address the above points in a final Stage 1 revision, in-principle acceptance should be forthcoming without requiring further in-depth Stage 1 review.

Response

Thank you very much for the thorough assessment of our revised manuscript. We do understand the concerns of reviewer 1 and took particular care in being completely explicit about the partial nature of the replication in the revised manuscript. We also addressed the concern regarding the inclusion criteria. Our point-by-point response to the reviewer' comments are detailed below.

Reviewer: 1

Thank you very much for taking the time to assess our revised Stage 1 manuscript. We address your concerns below. All changes made to the manuscript are marked in blue font colour in the revised manuscript.

Comments to the Author(s):

I would like to thank the authors for responding to my concerns.

I appreciate that the omission of the confidence ratings does not impact the other measures in the task (e.g., ToM, empathy). However, if the goal is to replicate the methods of the original study by Reiter et al. (2017) as closely as possible, this study has not achieved that with the current methodology. Meta-cognition is a key focus of the original study (discussed throughout the whole manuscript by Reiter et al., including the abstract) but it has been completely omitted here. Without the inclusion of this data, this study appears to only be a partial replication of Reiter et al. (2017). Perhaps this should be made more clear in the Introduction.

Response

Thank you for pointing this out. We are now very explicit and clear that we aimed to replicate the effect of age on empathy, compassion, and Theory of Mind observed in Reiter et al. (2017)'s study by editing the following sentences in the abstract and introduction:

"To this end we aimed to replicate **the effect of age on empathy, compassion and Theory of Mind a recent study by** observed in Reiter and colleagues (2017)'s study by using the same **naturalistic ecologically valid** paradigm **to investigate the effects of aging on socio-affective and socio-cognitive processes** in an independent sample with similar age ranges." (line 7 to 11; abstract)

"The reproducibility **of the results** of Reiter and colleagues' **results (59) on the effect of age on social affect and cognition** would have implications for the development of interventions aiming to improve healthy aging." (line 94 to 96; introduction)

"Therefore, the object of this study was to replicate **the effect of age on empathy, compassion and ToM found in** Reiter and colleagues' study (59) by using the same paradigm (EmpaToM) in a different setting (inside an MRI scanner) and **in** an independent sample of YA and OA but keeping all other methods as close as possible to the original study." (line 100 to 103; introduction)

We further added the following sentences in the method section (line 168 to 172):

"Other than Reiter and colleagues (59) we did not include a confidence rating after the multiple-choice questions, since we were especially interested in the effect of age on empathy, compassion, and ToM, rather than metacognition. Prior studies have shown that neither excluding the confidence rating at the end of the trial (37) nor replacing it by another rating (67) does affect the previous measures within each trial."

Comments to the Author(s):

I noticed there is a small difference in the reporting of the MoCA screening cut-off scores that the authors might want to clarify. The authors state: "Analogous to Reiter and colleagues, only OA with a score of 26 or higher were included in the current sample" but Reiter et al. reported that participants who scored below 25 on the MoCA were excluded. This would mean that people who scored 25 and above on the MoCA would have been included in Reiter et al.? I think it is fine if there is a difference but perhaps the authors should change their wording to make it clear that this was not the same cut off used in the original paper by Reiter et al.

Response

Thank you for making us aware of the difference in reporting the MoCa cut-off. We double-checked the cut-offs with the authors of Reiter et al. (2017). As you suggested there was a slight difference in the cut-offs. Reiter et al. (2017) used a cut-off of 25 and we used a cut-off of 26. Thus, Reiter et al. (2017) included participants with a score of 25 or higher on the MoCa and we included participants with a score of 26 or higher on the MoCa. We therefore changed the suggested sentence into (line 128 to 130):

"Analogous With a slight deviation from ~~to~~ Reiter and colleagues (59), who used a cut-off of 25, we used a cut-off of 26 on the MOCA (66). Hence, only OA with a score of 26 or higher on the MoCA were included in the current sample ~~(66)~~."

Reviewer: 4

Comments to the Author(s):

I am happy that the authors have addressed the comments of the reviewers.

Response

Thank you very much for taking the time to assess our revised Stage 1 manuscript. We are happy you are satisfied with our revision.

References used in this letter of response

Reiter, A. M. F., Kanske, P., Eppinger, B., & Li, S.-C. (2017). The Aging of the Social Mind—Differential Effects on Components of Social Understanding. *Scientific Reports*, 7(1), 11046. <https://doi.org/10.1038/s41598-017-10669-4>

References of the revised manuscript

37. Winter K, Spengler S, Bermpohl F, Singer T, Kanske P. Social cognition in aggressive offenders: Impaired empathy, but intact theory of mind. *Sci Rep*. 2017 06;7(1):670.
59. Reiter AMF, Kanske P, Eppinger B, Li S-C. The Aging of the Social Mind - Differential Effects on Components of Social Understanding. *Scientific Reports*. 2017 Sep 8;7(1):11046.
66. Davis DH, Creavin ST, Yip JL, Noel-Storr AH, Brayne C, Cullum S. Montreal Cognitive Assessment for the diagnosis of Alzheimer's disease and other dementias. *Cochrane Dementia and Cognitive Improvement Group, editor. Cochrane Database of Systematic Reviews [Internet]. 2015 Oct 29 [cited 2020 Dec 7]; Available from: <http://doi.wiley.com/10.1002/14651858.CD010775.pub2>*
67. Lehmann K, Böckler A, Klimecki O, Müller-Liebmann C, Kanske P. Feeling with or thinking as: How empathy and mentalizing shape prosociality. in prep.;

Appendix C

Dear Professor Chris Chambers,

We are delighted that our manuscript entitled “The Aging of the Social Mind: Replicating the preservation of socio-affective and the decline of socio-cognitive processes in old age” has been granted in-principle acceptance.

We have registered our approved protocol on the Open Science Framework (<https://doi.org/10.17605/OSF.IO/9BF3S>) and hereby submit our Stage 2 manuscript. Minor changes in wording for better understanding in the abstract, introduction or methods section are marked in blue font colour for easier identification in the Stage 2 manuscript.

We are looking forward to hearing from you.

Kind regards,

Julia Stietz and Philipp Kanske
On behalf of all authors

Appendix D

Dear Professor Chris Chambers,

We are delighted that two of the original Stage 1 reviewers returned to the evaluation of Stage 2 and recommended acceptance following a minor revision. Thank you very much for giving us the chance to revise our Stage 2 manuscript.

In the following, the reviewers' comments are in italics, whereas our responses are in standard font. In the revised manuscript, we marked all changes in blue font colour for easier identification.

We are looking forward to hearing from you.

Kind regards,

Julia Stietz and Philipp Kanske
On behalf of all authors

Associate Editor (Professor Chris Chambers):

Comments to the Author:

Two of the original Stage 1 reviewers kindly returned to evaluate the Stage 2 manuscript. As you will see, both are positive, finding that the primary Stage 2 criteria are met, and recommend acceptance following minor revision to correct typographic errors (Reviewer 4), and consider some minor addition to the Discussion (Reviewer 2). Based on my own reading, I feel the Discussion is sufficient as it stands for this article type, although I am happy for the authors to amend, if they wish, in accordance with the suggestion of Reviewer 2. Please respond to these points in a revision and full acceptance should be forthcoming without requiring further in-depth review.

Response

Thank you very much for the thorough assessment of our Stage 2 manuscript. We have carefully corrected the typographic errors pointed out by Reviewer 4, but have – as you suggested – not revised the discussion. Our point-by-point response to the reviewer' comments are detailed below.

Reviewer: 2

Thank you very much for returning and evaluating our Stage 2 manuscript.

Comments to the Author(s)

The results are interesting and mostly replicate those of the previous study. As they are relevant in the field of ToM in aging I would suggest commenting and elaborating them a little more in the discussion section.

Response

We appreciate your suggestions on commenting and elaborating the ToM results a little more in the discussion. We agree that the results are interesting and relevant to the field regarding the effect of age on ToM. However, in line with the Editor's note on this point, we decided not to amend the discussion here: as we aimed at replicating the findings observed by Reiter et al. (2017), we focused in our discussion on the replicability.

Reviewer: 4

Thank you very much for taking the time to evaluate our Stage 2 manuscript as well. We addressed your suggestions below. All changes made to the manuscript are marked in blue font colour in the revised manuscript

Comments to the Author(s)

There are just a handful of tyos:

On page 4, "believes" should be "beliefs"

On page 5, "That OA answered significant slower than YA..." should read "That OA answered significantly slower than YA..."

On page 9, "...an MRI-suitable keyboard that rested on there right thigh..." should read "...an MRI-suitable keyboard that rested on their right thigh..."

On page 9, "This speaks against the notion proposed by several authors (84,85) that age difference in perspective taking..." should read "This speaks against the notion proposed by several authors (84,85) that age differences in perspective taking..."

Response

Thank you very much for pointing out these typographic errors. We carefully corrected them in the manuscript.

Line 41-43: "Some researchers also distinguish between an affective and cognitive component of ToM, with the former being the reasoning about emotions and the later one the reasoning about thoughts and beliefs (25)."

Line 176-177: "That OA answered significantly slower than YA on the IDP is in line with Reiter and colleagues' sample (62) as well as findings of a population-based lifespan sample (74)."

Line 332-335: "Reiter and colleagues (62) had participants sit in front of a computer answering the questions on a regular keyboard, whereas our participants lay in an MRI scanner answering the questions on an MRI-suitable keyboard that rested on their right thigh with their fingers directly on the keys."

Line 359-361: "This speaks against the notion proposed by several authors (84,85) that age differences in perspective taking are mediated through differing educational levels in these cohorts."

References used in this letter of response

Reiter, A. M. F., Kanske, P., Eppinger, B., & Li, S.-C. (2017). The Aging of the Social Mind—Differential Effects on Components of Social Understanding. *Scientific Reports*, 7(1), 11046. <https://doi.org/10.1038/s41598-017-10669-4>

References of the revised manuscript

25. Healey ML, Grossman M. Cognitive and Affective Perspective-Taking: Evidence for Shared and Dissociable Anatomical Substrates. *Front Neurol* [Internet]. 2018 Jun 25 [cited 2021 Jun 15];9:491. Available from: <https://www.frontiersin.org/articles/10.3389/fneur.2018.00491/full> DOI: 10.3389/fneur.2018.00491
62. Reiter AMF, Kanske P, Eppinger B, Li S-C. The Aging of the Social Mind - Differential Effects on Components of Social Understanding. *Scientific Reports* [Internet]. 2017 Sep 8 [cited 2018 Sep 25];7(1):11046. Available from: <https://www.nature.com/articles/s41598-017-10669-4> DOI: 10.1038/s41598-017-10669-4
74. Lindenberger U, Mayr U, Kliegl R. Speed and intelligence in old age. *Psychology and Aging* [Internet]. 1993 [cited 2021 Jun 15];8(2):207–20. Available from: <https://psycnet.apa.org/doiLanding?doi=10.1037%2F0882-7974.8.2.207> DOI: 10.1037/0882-7974.8.2.207
84. Riva F, Triscoli C, Lamm C, Carnaghi A, Silani G. Emotional Egocentricity Bias Across the Life-Span. *Front Aging Neurosci* [Internet]. 2016 [cited 2021 Jun 15];8. Available from: <https://www.frontiersin.org/articles/10.3389/fnagi.2016.00074/full> DOI: 10.3389/fnagi.2016.00074
85. Li X, Wang K, Wang F, Tao Q, Xie Y, Cheng Q. Aging of theory of mind: The influence of educational level and cognitive processing. *International Journal of Psychology* [Internet]. 2013 [cited 2021 Jun 15];48(4):715–27. Available from: <https://onlinelibrary.wiley.com/doi/abs/10.1080/00207594.2012.673724> DOI: 10.1080/00207594.2012.673724